# Nepal Ambient Monitoring and Source Testing Experiment (NAMaSTE): Emissions of particulate matter and sulfur dioxide from vehicles and brick kilns and their impacts on air quality in the Kathmandu Valley, Nepal

Min Zhong[1,2], Eri Saikawa[1,3], Alexander Avramov[1], Chen Chen[1], Boya Sun[1], Wenlu Ye[3], William C. Keene[4], Robert J. Yokelson[5], Thilina Jayarathne[6,7], Elizabeth A. Stone[6], Maheswar Rupakheti[8], and Arnico K. Panday[9]

[1]Department of Environmental Sciences, Emory University, Atlanta, GA
[2]Now at Department of Environmental Engineering, Texas AM University-Kingsville, Kingsville, TX
[3]Rollins School of Public Health, Emory University, Atlanta, GA
[4]Department of Environmental Sciences, University of Virginia, Charlottesville, VA
[5]Department of Chemistry, University of Montana, Missoula, MT
[6]Department of Chemistry, University of Iowa, Iowa City, IA
[7]Now at Department of Chemistry, Purdue University, West Lafayette, IN
[8]Institute for Advanced Sustainability Studies, Potsdam, Germany
[9]International Centre for Integrated Mountain Development (ICIMOD), Khumaltar, Nepal

**Correspondence:** Eri Saikawa (eri.saikawa@emory.edu)

**Abstract.** Air pollution is one of the most pressing environmental issues in the Kathmandu Valley, where the capital city of Nepal is located. We estimated emissions from two of the major source types in the valley (vehicles and brick kilns) and analyzed the corresponding impacts on regional air quality. First, we estimated the on-road vehicle emissions in the valley using the International Vehicle Emission (IVE) model with local emission factors and the latest available data for vehicle registration. We also identified the locations of the brick kilns in the Kathmandu Valley and developed an emissions inventory for these kilns using emission factors measured during the Nepal Ambient Monitoring and Source Testing Experiment (NAMaSTE) field campaign in April 2015. Our results indicate that the commonly used global emissions inventory, the Hemispheric Transport of Air Pollution (HTAP_v2.2), underestimates particulate matter emissions from vehicles in the Kathmandu Valley by a factor greater than 100. HTAP_v2.2 does not include brick sector and we found that our sulfur dioxide ($SO_2$) emissions estimates from brick kilns are comparable to 70% of the total $SO_2$ emissions considered in HTAP_v2.2. Next, we simulated air quality using the Weather Research and Forecasting model coupled with Chemistry (WRF-Chem) for April 2015 based on three different emission scenarios: HTAP only, HTAP with updated vehicle emissions, and HTAP with both updated vehicle and brick kilns emissions. Comparisons between simulated results and observations indicate that the model underestimates observed surface elemental carbon (EC) and $SO_2$ concentrations under all emissions scenarios. However, our updated estimates of vehicle emissions significantly reduced model bias for EC, while updated emissions from brick kilns improved model performance in simulating $SO_2$. These results highlight the importance of improving local emissions estimates for air quality modeling. We further find that model overestimation of surface wind leads to underestimated air pollutant concentrations in the Kathmandu

Valley. Future work should focus on improving local emissions estimates for other major and underrepresented sources (e.g., crop residue burning and garbage burning) with a high spatial resolution, as well as the model's boundary-layer representation, to capture strong spatial gradients of air pollutant concentrations.

## 1 Introduction

Air pollution is one of the most pressing environmental issues in South Asia. According to the 2016 Environmental Performance Index, air quality in Nepal ranked fourth worst in the world (Angel and Alisa, 2016) and Kathmandu, its capital and largest metropolis, is one of the most polluted cities in Asia. Kathmandu lies in a bowl-shaped valley with a floor elevation of $\sim$1300 m surrounded by mountains of 2000 to 2800 m. It is inhabited by approximately 2.5 million people with a steady population growth of 4% $yr^{-1}$ (Muzzini and Aparicio, 2013). The primary sources of air pollution within the valley are uncontrolled emissions from vehicles, brick kilns, and biomass and garbage combustion coupled with dust originating from both local fugitive emissions and long-distance transport (Gronskei et al., 1996; Kim et al., 2015; Shakya et al., 2010; Stone et al., 2010, 2012). The unique topography coupled with high emissions of pollutants contribute to low air quality in the valley.

The rapid growth of the vehicle fleet is of particular concern. According to the Department of Transport Management (DoTM) in Nepal, the total number of registered vehicles in the Bagmati Zone (most of which operate in the Kathmandu Valley) increased from 292,697 to 922,831 (12% $yr^{-1}$), between 2005 and 2015 (DoTM, 2017). Approximately 80% of the total registered vehicles are motorcycles, while cars and pickup trucks account for another 13%. These vehicles emit air pollutants, including carbon monoxide (CO), nitrogen oxides ($NO_x = NO + NO_2$), non-methane volatile organic compounds (NMVOCs), particulate matter (PM), elemental carbon (EC), and organic carbon (OC). Buses were estimated to emit more than 90% of OC, EC, and $NO_x$ among all vehicles, while motorcycles were estimated to emit large amounts of CO (50%) and NMVOCs (66%) by Shrestha et al. (2013). In 2010, the annual emissions of EC and OC from colorreddiesel-powered vehicles in the Kathmandu Valley were estimated at 2,117 and 570 ton/year, respectively (Shrestha et al., 2013). Relative to estimates for the Kathmandu Valley in 2010 based on the global emissions inventory Hemispheric Transport of Air Pollution (HTAP_v2.2; Janssens-Maenhout et al. (2015)), the above estimates for vehicle emissions of EC and OC are 80 and 20 times higher, respectively, than those estimated for all sectors combined in HTAP. Considering that Shrestha et al. (2013) did not include emissions from personal cars or trucks in their estimates, vehicle emissions in the Kathmandu Valley appear to be significantly underestimated in HTAP_v2.2.

More than 100 brick kilns of different types throughout the Kathmandu Valley produce over 600 million bricks per year (Quest Forum Pvt. Ltd, 2017; Gronskei et al., 1996; Weyant et al., 2014). The majority of these kilns (97%) use the Fixed Chimney Bull's Trench Kiln (FCBTK) technology and its variations such as zigzag. In the Kathmandu Valley, brick kilns typically operate for six months a year, generally from December to May. The main fuels burned in brick kilns are biomass and high-sulfur coal (Joshi and Dudani, 2008), both of which emit $SO_2$ and PM but in differing amounts. Burning biomass emits relatively greater amounts of PM, whereas coal combustion emits relatively greater amounts of $SO_2$ (Stockwell et al., 2016; Jayarathne et al., 2018). Weyant et al. (2014) estimate the total emissions from the brick industry in South Asia to be 120 Tg

yr$^{-1}$ carbon dioxide, 2.5 Tg $^{-1}$ CO, 0.19 Tg $^{-1}$ PM$_{2.5}$ (PM with an aerodynamic diameter less than 2.5 $\mu$m) and 0.12 Tg $^{-1}$ BC. Pariyar et al. (2013) report that brick kilns contribute more than 60% of total SO$_2$ and PM emissions in the Kathmandu Valley. Using a source apportionment method, Kim et al. (2015) found that brick kilns contribute 40% of EC concentrations in the Kathmandu Valley in winter. Despite being one of the major sources of air pollution, brick kiln emissions are not included

in the existing gridded inventory estimates. The developers of the Regional Emission inventory in ASia version 2 (REAS v2; Kurokawa et al. (2013)) discussed that brick kilns are one of the major sources of EC, but their inventory did not include emissions from this sector. HTAP_v2.2 uses REAS v2 for Nepal and thus does not include emissions from brick kilns in their inventory either. Due to the lack of gridded emissions estimates, to date, no studies have explicitly simulated the impacts of brick kiln emissions on regional air quality.

The purpose of this study was to analyze the emissions and air quality impacts due to on-road vehicles and brick kilns in the Kathmandu Valley. Mues et al. (2018) reported that an emission database is essential to improve the simulation of EC using regional chemical transport models in the Kathmandu region. We first estimated on-road traffic emissions using the latest number of registered vehicles and emission factors generated for local conditions. We also created a point-source emissions inventory for brick kilns using the newly-measured emission factors in the Kathmandu Valley. In the recent Nepal Ambient Monitor-

ing and Source Testing Experiment (NAMaSTE) in April 2015, *in situ* emissions from several important under-characterized combustion emission sources were measured, including brick kilns and motorcycles (Jayarathne et al., 2018; Stockwell et al., 2016). Emission factors obtained from NAMaSTE were used to create the point source emission inventory of brick kilns. We then modified the HTAP_v2.2 estimates with updated emissions from vehicles and brick kilns. Next, we conducted three simulations using Weather Research and Forecasting model coupled with Chemistry (WRF-Chem) to explore the impacts of

emissions from vehicles and brick kilns on the local air quality. These three simulations differed only in emission scenarios for vehicles and brick kilns; all other model conditions were kept the same.

## 2    Method

### 2.1    WRF-Chem model description

We used the regional chemical transport model WRF-Chem version 3.5 in this study. The Regional Atmospheric Chemistry

Mechanism (RACM) (Stockwell et al., 1997) is used for gas-phase reactions. Aerosol chemistry is represented by the Model Aerosol Dynamics for Europe with the Secondary Organic Aerosol Model (MADE/SORGAM) (Ackermann et al., 1998; Schell et al., 2001). MADE/SORGAM predicts the mass of several particulate-phase species, including sulfate, ammonium, nitrate, sea salt, dust, EC, OC, and secondary organic aerosols in the three aerosol modes (Aitken, accumulation, and coarse). This aerosol model has been widely used in previous studies (e.g., Gao et al. (2014); Kumar et al. (2012); Saikawa et al.

(2011); Tuccella et al. (2012); Zhong et al. (2016)). Photolysis rates are based on the Fast-J photolysis scheme (Wild et al., 2000). The Rapid Radiative Transfer Model (RRTM) (Mlawer et al., 1997) accounts for aerosol-radiative feedbacks. Lin et al. (1983) and one-and-a-half local Mellor-Yamada-Nakanishi-Niino Level 2.5 (Nakanishi and Niino, 2006) schemes are used to parameterize cloud microphysical and sub-grid processes in the planetary boundary layer (PBL), respectively. The horizontal

winds, temperature, and moisture at all vertical levels are nudged to the large-scale meteorological fields from the National Center for Environmental Prediction (NCEP) Global Forecast System final gridded analysis datasets.

The model domain covered large parts of the Himalayas, India, Nepal, and Southwest China (Fig. 1). The domain included three levels of one-way nesting with horizontal grid spacing of 27, 9, and 3 km each of which was centered on the Kathmandu Valley. The topography of the innermost model domain is complicated, with the Himalayas range sitting across west to east and separating the Indian subcontinent from the Tibetan Plateau. Even when we use 3 km spacing for the nested domain, the model is unable to resolve the very steep topographic features but this was the best we could do with this project, given the resolution of emissions available. There are 31 vertical levels from the surface to 50 mb. The Model for OZone And Related Transport (MOZART) global chemical transport model (Emmons et al., 2010) was used to provide initial and lateral boundary conditions for chemical species in the outer most domain. We found that the inclusion of dust from MOZART led to overestimated aerosol optical depth (AOD) at Jomsom in Nepal and at Qomolangma (Mt. Everest) station for Atmospheric and Environmental Observation and Research, Chinese Academy of Sciences (QOMS_CAS) in Tibet, China. Therefore, in our simulations, dust concentrations were calculated online, using the Air Force Weather Agency (AFWA) emission scheme (Marticorena and Bergametti 1995) with zero initial conditions.

To analyze the impact of emissions from vehicles and brick kilns on air quality in the Kathmandu Valley, we performed a set of three nested model simulations, referred to as HTAP, HTAP_vehicle, and HTAP_vehicle_brick. The HTAP simulation used the original HTAP_v2.2 emission inventory as inputs. The HTAP_vehicle simulation used HTAP_v2.2 with updated vehicle emissions (Section 2.2.1) as inputs. The HTAP_vehicle_brick simulation, used the HTAP_v2.2 with updated vehicle emissions and the additional brick kiln emissions (Section 2.2.2) as inputs. The same meteorology and boundary inputs were used for all three nesting simulations as our focus is to understand the impact of emissions on the local air quality. We conducted each simulation for the two week period of April 12-24, 2015 during which observational data from the NAMaSTE field campaign were available for comparison (Section 2.3) . The model was spun-up for 5 days preceding the simulation period, which was sufficient to ventilate the regional domain.

## 2.2 Emissions

### 2.2.1 Emission Scenarios in 2015

We used three emissions scenarios (Table 1) to investigate the impact of emissions on local air quality in the Kathmandu Valley. The first emissions scenario is the same as the original HTAP_v2.2 Janssens-Maenhout et al. (2015). HTAP is a gridded global emission inventory combined with the regional inventories and gap-filled with the Emissions Database for Global Atmospheric Research (EDGAR v4.3) (Janssens-Maenhout et al., 2013). In Asia, HTAP_v2.2 uses MIX inventory, a regional emission inventory in Asia, which is also developed based on the 'mosaic' approach including multiple existing national inventories (Li et al., 2017). The second emissions scenario utilizes the original HTAP_v2.2 with updated vehicle emissions (Section 2.2.2). The third scenario is built on the second scenario and adding emissions from brick kilns (Section 2.2.3). We used the latest

available HTAP_v2.2 for 2010 as the baseline inventory, as this is the closest year to 2015 that we have the data for. The vehicle and brick kiln emissions were developed for year 2015.

### 2.2.2 Emissions from vehicles

Emissions from the road transport sector in the Kathmandu Valley were estimated using the International Vehicle Emission (IVE) model version 2.0 (Davis et al., 2005). The IVE model is specifically designed to calculate emissions from motor vehicles in developing countries for local conditions and has been used extensively in several countries worldwide, including Nepal (Shrestha et al., 2013), India (Barth et al., 2007), China (Guo et al., 2007; Wang et al., 2008), and Iran (Shahbazi et al., 2016). Emissions were estimated by the product of the base emission factors, the correction factors, and the distance travelled or the total starts for each of the vehicle categories. We classified vehicles into six overall categories (motorcycles, buses, cars, trucks, taxis, and 3-wheelers), numerous technology-based subcategories, and two fuel types (gasoline and diesel). For fuel quality input values, we used unleaded gasoline with a sulfur content of 300 ppm and diesel with a sulfur content of 500 ppm. The base emission factors are derived from emissions tests conducted mainly in the USA, along with data collected in developing countries. The correction factors consider local conditions (meteorology, altitude, inspection/maintenance program, etc.), fuel quality, and power and driving characteristics (vehicle specific power pattern, road grade, air conditioning usage, and start pattern). Local meteorology data, such as ambient temperature and relative humidity were obtained from Weather Underground for April 2015. Driving characteristics for motorcycles, buses, taxis, and 3-wheelers were adopted from surveys in the Kathmandu Valley, documented by Shrestha et al. (2013). Since we lack survey data for trucks and cars in Kathmandu, we used the data from Pune, India for these two types of vehicles (Barth et al., 2007). Pune was the only representative city within South Asia, where the International Sustainable System Research Center (ISSRC) conducted a detailed study of vehicle activity. The travel distances were obtained from the vehicle registration number and vehicle kilometers traveled (VKT). Daily VKT and the number of starts per day were adopted from Shrestha et al. (2013), which is specific for vehicles in the Kathmandu Valley. The number of vehicles in 2015 was taken from the government report of the DoTM (2017). Data used for the number of vehicles and VKT for each vehicle category in 2015 are summarized in Table 2. Table S1 lists the detailed technology of each category, fraction of vehicles with different technology in fleet, and corresponding European vehicle emission standards (Euro Standards).

We used the IVE model to estimate emissions of CO, $NO_x$, NMVOC, $SO_2$, PM, and some greenhouse gases. All emitted PM was assumed to be $PM_{2.5}$ because the ratio of $PM_{2.5}$ to $PM_{10}$ is 0.92 for diesel vehicles and 0.88 for gasoline vehicles in EPA 2014 MOVES model. Studies such as Gillies et al. (2001); Handler et al. (2008) have also found that 74% and 67% of $PM_{10}$ is $PM_{2.5}$ in on-road studies. Although we understand that assuming all emitted $PM_{10}$ to be $PM_{2.5}$ is potentially an overestimation, we believe that this is acceptable, given the lack of observational data in Nepal or in South Asia.

Because the IVE model does not directly estimate emissions of EC or OC, we used conversion factors derived from the study of Kim Oanh et al. (2010) to estimate these emissions. Kim Oanh et al. (2010) specifically focused on the emissions of diesel vehicles in developing countries and had tested a large number of vehicles. For vans, we used EC/PM mass ratio of 0.46 and OC/PM of 0.2, while for trucks and buses, we used EC/PM of 0.48 and OC/PM of 0.13. We collected a group of

these conversion factors from different studies in Table S-3. Our EC/PM mass ratio is close to median value of all the studies listed below. The total vehicle emissions from IVE were distributed spatially and temporally based on HTAP_v2.2. While we acknowledge that using conversion factors from one study ignores the potential uncertainty due to driving pattern, weather conditions, fuel quality, and vehicle characteristics, we also feel that our estimate provides a good middle ground, given the existing study results.

### 2.2.3 Brick kiln emissions

We identified the kiln types in the Kathmandu Valley by comparing the specific images of brick kiln types with Google Earth Images dated April 2015. The monthly emissions of compound $i$ ($E_{i,j}$, g month$^{-1}$) for a certain type of brick $j$ were calculated as the product of the amount of fuel consumed ($BK_j$, kg-fuel) and the corresponding emission factor ($EF_{i,j}$, g kg-fuel$^{-1}$):

$$E_{i,j} = BK_j \times EF_{i,j} \tag{1}$$

The calculated emissions were mapped based on the location of each kiln.

The amount of fuel burned for each brick kiln $BK_j$ is estimated using the following formula:

$$BK_j = P_j \times W_{brick} \times E_{brick}/U_{fuel} \tag{2}$$

where $P_j$ is the production of brick kiln $j$ (number of bricks produced per month per kiln), $W_{brick}$ is the average weight of a brick (kg brick$^{-1}$), $E_{brick}$ is the specific energy consumption of a brick (MJ kg$^{-1}$), and Ufuel is the specific energy density of fuel (MJ kg-fuel$^{-1}$). Since the production of each brick kiln was not available, we estimated the monthly mean production using one-sixth of the annual average production, as brick kilns in the Kathmandu Valley usually operate six months a year. The annual mean production was obtained from a report submitted to the government of Nepal (SMS Environment and Engineering Pvt. Ltd, 2017). Most brick kilns in the Kathmandu Valley are fueled by high-sulfur coal (70%), which is supplemented with sawdust (24%), and wood and other fuels (6%) (Joshi and Dudani, 2008). Considering that the dominant fuel for brick firing is coal and that the EFs used here correspond to emissions from coal-fueled brick kilns, we used the specific energy density of coal to estimate emissions. Values and references for each variable in Eq. 2 are presented in the Supporting Information (Table S-2).

EF values measured from a zigzag kiln during the NAMaSTE field campaign are applied in Eq. 1 to calculate emission estimates of various trace gases and PM. Table S-1b lists the species, associated EF, and references that we estimated the emissions for. We used EFs of a zigzag kiln rather than a clamp kiln for all types of kilns in the valley because zigzag is the most common kiln type identified in the valley. We will provide more accurate emission estimates in our emission inventory when EFs of different types of kilns become available.

### 2.2.4 Other emissions

Anthropogenic emissions of other gaseous pollutants (CO, NO$_x$, NH$_3$, SO$_2$, and NMVOCs) and PM (EC, OC, PM$_{2.5}$, and PM$_{10}$) from major sectors are taken from HTAP_v2.2 for 2010. HTAP_v2.2 is the most recent global emissions inventory that

includes emissions from various sectors such as energy, industry, agriculture, residential (including both heating and cooking), aircraft, and shipping and has the highest spatial resolution. However, as mentioned earlier, HTAP does not currently include brick kiln emissions in their estimates and it also excludes large-scale biomass burning and crop residue burning. For non-residential "open" biomass burning emissions, we therefore used emissions from the Fire INventory from NCAR (FINN) inventory for the year 2015 (Wiedinmyer et al., 2011). For biogenic emissions, we used the Model of Emissions of Gases and Aerosols from Nature (MEGAN) version 2.1 (Guenther et al., 2012). Dust emissions are calculated online, using the AFWA emission scheme, as described above.

## 2.3   Observations and statistical methods for comparisons

We compared our simulations with the surface observations of air temperature, relative humidity, and wind speed at two sites in the Kathmandu Valley. The meteorological data at the Bode site at a height of 23 m were collected during the NAMaSTE field campaign and the data at the Tribhuvan International Airport site at standard meteorological monitoring heights (2 m for air temperature and relative humidity, and 10 m for wind speed and direction) were provided by the Department of Hydrology and Meteorology of the Ministry of Population and Environment of the Government of Nepal. The Bode site is located in the eastern part of the Kathmandu Valley at latitude of 27.689 °N and longitude of 85.395 °E. The altitude is about 1337 m. The airport is approximately 4 km west of Bode (Fig. 2), at approximately the same altitude.

The daily ground-based AOD values at 550 nm were obtained from the Aerosol Robotic Network (AERONET). We used Level 2.0 for the QOMS_CAS site in China and the Level 1.5 for the Jomsom site in Nepal. In addition, we also compared the space-based AOD values retrieved from the MODerate Resolution Imaging Spectrometer (MODIS) instrument aboard the Terra satellite with the simulated AOD from WRF-Chem. MODIS provides AOD retrievals at a resolution of $10\times10$ km. In this study, we used Level 2 and Collection 6 aerosol optical thickness at 550 nm. Concentrations of EC and $SO_2$ at Bode, Kathmandu were sampled at a height of 20 m during the NAMaSTE field campaign in April 12-24, 2015. We also compared simulated and observed surface $SO_2$ concentrations at several other sites in the valley (Kiros et al., 2016). The observed surface $SO_2$ is 8-week mean concentrations between March 23 and May 18, 2013 from Kiros et al. (2016). They were measured at 15 sites in the valley, including five urban sites (Bode, Indrachowk, Maharajgunj, Mangal Bazaar, Suryabinayak ), four suburban sites (Bhaisepati, Budhanilkantha, Kirtipur, Lubhu), and six rural sites (Bhimdhunga, Nagarkot, Naikhandi, Nala Pass, Sankhu, Tinpiple) (Kiros et al., 2016).

The overall performance of WRF-Chem in simulating meteorological data and air pollutants against observations was evaluated using the correlation coefficient ($r$), the normalized mean bias (NMB), the mean fractional bias (MFB), the mean fractional error (MFE), and the root mean square error (RMSE). The evaluation is based on bi-week statistics using the daily mean values weighted for the day and night sampling times at each site.

## 3 Emissions comparison

### 3.1 Vehicle emissions

The numbers, mileage, and starts for different vehicle types in the Kathmandu Valley, based on the vehicle registration information, is summarized in Table 2. Using the IVE model, we estimated monthly vehicle emissions for CO, $SO_2$, $NO_x$, NMVOCs,
EC, OC, and $PM_{2.5}$ (Table 3). Relative to the vehicle emissions estimates for 2010 by Shrestha et al. (2013), our estimates are about 2 to 4 times higher due to: (1) the increases in numbers of vehicles between 2010 and 2015; and (2) the inclusion of diesel trucks that were not considered in the earlier study. The total number of vehicles in this study is about 70% higher than that of Shrestha et al. (2013). In addition, the estimated running EFs for trucks are the highest among seven categories of vehicles for all air pollutants, with values 4 to 5 times higher than those for buses, which ranked the second highest (Table S0
and S1). Although trucks account for only 2.3% of the total numbers of vehicles, they are the major contributor to pollutant emissions due to their substantially higher EFs. Trucks account for more than 80% of monthly total emissions for both $PM_{2.5}$ and $NO_x$, and 50% or more for the other pollutants.

Comparison of these new emissions estimates with those from the ground transport sector in the HTAP_v2.2 emissions inventory reveal that CO, $NO_x$, NMVOCs, EC, OC and $PM_{2.5}$ are significantly underestimated in HTAP (Table 3). For example,
the emission estimates of $PM_{2.5}$, OC and EC calculated using the IVE model were factors of 186, 100, and 375, respectively, greater than those in HTAP. In contrast, $SO_2$ emissions estimates agreed well and only differed by 17%. Our revised emissions of $PM_{2.5}$, EC, CO, $NO_x$ from vehicles drive the substantially greater total emissions of these species in the Bagmati Zone relative to those based on the HTAP (Fig. 3).

### 3.2 Brick kiln emissions

We found 112 brick kilns in the Kathmandu Valley, consistent with a previously reported total of 110 (SMS Environment and
Engineering Pvt. Ltd, 2017). Fig. 2 shows the spatial distribution of these brick kilns. Approximately 40% of brick kilns are located in the southern portion of the valley, 35% in the eastern portion, and the rest are in the western portion. We identified four types of brick kilns, including FCBTK, Hoffman kiln, Vertical Shaft Brick Kiln (VSBK), and zigzag kiln. Out of the 112 kilns, the dominant types were zigzag (63) and FCBTK (46), while there were only three Hoffman kilns and VSBK kilns
combined. Based on Eq. (2), the average fuel consumption of these kilns was about 9,700 ton/month or 58,200 ton/year, which is close to 65,100 ton/year estimated in a previous report (SMS Environment and Engineering Pvt. Ltd, 2017).

Table 3 summarizes the estimated monthly total emissions of major air pollutants from brick kilns in the Kathmandu Valley. Of these species, those with the greatest mass emitted were $PM_{2.5}$ (135 ton/month), followed by $SO_2$ (123 ton/month) and CO (98 ton/month), while emissions for other pollutants were less than 20 ton/month. Table 3 also compares emissions from brick
kilns and those from all other sectors in the HTAP_v2.2 emissions inventory. Our brick kiln $SO_2$ and $PM_{2.5}$ emissions estimates are each equivalent to 68% and 16%, respectively, of the total emissions in the HTAP estimates. The increase in $PM_{2.5}$ and $SO_2$ emissions due to adding the brick kiln emissions can be seen clearly in (Fig. 3). For EC, OC, CO, NOx, and NMVOCs, the brick kiln sector contribution is less than 3% of our updated total emissions.

## 4 Model results and evaluation

### 4.1 Meteorology

The different emission scenarios did not impact simulated meteorological conditions. Therefore, we only present the statistical analysis of our HTAP_vehicle_brick simulation, using the HTAP inventory with updated emissions for both vehicles and brick kilns. Model simulated 2-m temperature and relative humidity, as well as 10-m wind speed, are compared to observations at two sites in the Kathmandu Valley: Bode and the Tribhuvan International Airport. Fig. 4 shows the comparisons of predicted daily-averaged quantities with observations and Table 4 presents the statistical indices of comparisons for each site. The temperature at Bode is simulated with a correlation of 0.8 and a small negative NMB of 4.7%. At the airport, the correlation of 0.7 is close to that at Bode, but a larger bias is observed, with NMB of 15.8%. The model systematically underestimates relative humidity with a correlation of 0.5-0.6 and NMB of -40.8% to -34.0%, due to an underestimation of both minima and maxima. In a previous WRF-Chem study in the Kathmandu Valley (Mues et al., 2018), an underestimation of relative humidity was also clearly observed near the ground. The temporal correlation coefficient of daily 10-m wind speed is 0.7 at the airport and 0.8 at Bode. Although the model reproduces the daily variability well, it overestimates the wind speed at both sites. The model performs better at Bode (NMB = 67%) than at the airport site (NMB = 176%) in simulating wind speed. The modeled mean wind speed at the airport site is about 1.68 m s$^{-1}$ higher than the observation, with RMSE of 1.74 m s$^{-1}$. WRF-Chem usually has difficulty simulating wind speed over complex mountain terrains; a larger bias over mountain regions was also found in previous studies (Mar et al., 2016; Mues et al., 2018). Zhang et al. (2013) explained that the overestimation in wind speeds is likely caused by poor representation of surface drag exerted by unresolved topographical features in WRF-Chem. A closer look at the wind observational data reveals the presence of a local, thermally-driven diurnal wind circulation that controls the airflow regime in the Kathmandu Valley during weak gradient synoptic-scale flow and arguably, significantly impacts the air quality in the valley (Fig. 5). During the day, the winds at both Bode and the airport are predominantly from the SW quadrant along the axis of the nearby river, with hourly-averaged magnitudes of up to 5 m s$^{-1}$. In contrast, the katabatic winds during the night are much weaker, generally under 1 m s$^{-1}$ and with prevailing easterly component. Note that winds at Bode are consistently stronger than those at the airport. One of the most likely reasons for that is the differing measurement height at both sites: 23 m above surface at Bode and 10 m at the airport.

The model generally reproduces the wind direction shift at both sites quite well (Figs. 6 and 7); however, the wind speed magnitude is substantially overpredicted. The overestimation is not that large during the day but it is severe during the nighttime hours, suggesting serious differences in the structure of simulated and observed nighttime boundary layers. We should note here that the model does not directly predict the wind field at 10-m height; instead it is extrapolated from the first model level using Monin-Obukhov similarity theory (Jimenez et al, 2012), and therefore is highly influenced by the PBL scheme used in the simulations.

## 4.2 AOD

AOD is a column-integrated measurement and, thus, not directly correlated with concentrations of near-surface PM. However, in the context of model validation, AOD is useful as a general indicator of near-surface air quality. Fig. 8 depicts the spatial pattern of the 2-week mean AOD observed by MODIS and the modeled AOD at 550 nm wavelength. In general, the simulated AOD values are higher in the southern part of the domain than those in the northern part. MODIS AOD also shows a similar spatial distribution in the southern part but most data in the northern part are missing due to cloud coverage. It is clear from the figure that adding brick kiln emissions has little impact on AOD in the Kathmandu Valley, while modifying vehicle emissions leads to a significant increase in modeled AOD. The average difference in AOD between the simulations with and without the revised vehicle emissions is 18% in the Kathmandu Valley. As discussed in section 3.2, diesel engines emit a large amount of EC. These aerosols strongly absorb sunlight at all UV-Vis wavelengths and, consequently, contribute to higher AOD values.

The time series of simulated versus observed daily mean AOD at the two AERONET sites within the model domain (Jomsom, Nepal and QOMS_CAS, China) and the corresponding performance statistics are presented in Fig. 9. The model tends to overestimate the lower observed values at the QOMS_CAS site, with an MFB of 56%. At Jomsom, the model predicts the lower measured AOD values reasonably well during April 15 to 24. However, the model misses the peak on April 14, when the observed AOD is near 1.0, an indication of severe air pollution. Instead, our model predicts a somewhat lower peak on the preceding day. This high AOD value was driven by emissions from a wildfire located southeast of Jomsom. It can be seen clearly from the model simulation (Fig. S1) that the fire caused high surface PM and CO concentrations near the burning area on April 12. Since the prevailing wind direction on April 13 in the simulation was from the southeast and with the overestimated wind speed, the smoke was transported to northwest and increased the simulated AOD value at the Jomsom site earlier than observed.

## 4.3 EC

Fig. 10 shows the spatial pattern of simulated EC, averaged during the simulation period. The average simulated EC concentration in the Kathmandu Valley during the two-week simulation period was approximately 6.2 $\mu$g m$^{-3}$, higher than concentrations in the surrounding regions. Vehicles (primarily diesel trucks and buses) contribute approximate 85% of total EC emissions, whereas brick kilns account for only about 0.11%. Consequently, the simulation that includes brick kiln emissions (HTAP_vehicle_brick) does not improve the model performance in predicting EC at Bode relative to that with updated vehicle emissions (HTAP_vehicle).

Fig. 11 depicts the time series of observed and simulated surface EC concentrations. The average EC concentration observed at the Bode during the campaign was 5.6 $\mu$g m$^3$ during daytime, 10.82 $\mu$g m$^3$ during night time, and 8.32 $\mu$g m$^3$ for the 24-hour average. Two factors contributed to the nighttime increase in surface concentrations: 1) the diminishing mixing-layer depth and 2) the air flow circulation shift in response to surface cooling. As the night progresses, the turbulent mixing in the developing nocturnal boundary layer is suppressed and air pollutants are confined in a shallow layer close to surface. The shift in the wind direction is also conducive to increased surface concentrations as the Bode site is located downwind with respect

to the major cluster of brick kilns in the area (Figs. 2 and 5). In contrast, during daytime the mixing-layer depth increases in response to solar heating and promotes the vertical mixing throughout the depth of the whole boundary layer, leading to a notable decrease in surface concentrations. The Bode site during the day also is upwind from the brick kiln cluster, which leads to further reduced concentrations. This idealized scenario holds true for most of the simulation period with the exception of the days between April 13 and 16, when the local air flow circulation is disrupted by a large-scale disturbance. During that period, the wind speeds are generally below 2 m s$^{-1}$ even during the day, suggesting that the peak in the surface concentrations (Fig. 11) could be related to suppressed boundary-layer mixing.

Observed EC concentrations are strongly underestimated with poor correlation when the original HTAP emissions are used as model input (MFB = -125%, $r$ = 0.19 for 24-hour averaged EC). This is similar to Mues et al. (2018), who reported an underestimation of EC concentrations by a factor of five when using HTAP emissions in their simulations. The simulation with updated vehicle emissions (HTAP_vehicle) shows reduced bias and better correlation (MFB = -73%, $r$ = 0.61 for 24-h average), although the model still underestimates EC concentrations.

The model captures the daytime low concentrations very well during April 18-21, but it fails to predict the observed peak on April 15-16. Such high EC concentration episodes can be either caused by stagnant meteorological conditions or enhanced emissions. We examined the precipitation and wind speed at Bode during the high and low episode periods (Table S2). On April 16 (high EC) and 19 (low EC), no rainfall was observed. The daytime wind speed on April 16 (2.4 m s$^{-1}$) was slightly lower than that on April 19 (2.7 m $^{-1}$) but the EC concentration on April 16 was six times higher. This suggests that a sporadic emission source (e.g. garbage or biomass burning) that was not accounted for in our model could be responsible for the observed high concentration episode.

The nighttime surface concentrations are significantly underestimated by the model throughout the entire period. One possible cause for this underestimation is illustrated in Fig. S2, showing the diurnal evolution of the modeled mixing layer height. During the night, the simulated boundary layer remains well mixed up to a height of 500 m, which facilitates the vertical transport of pollutants, and consequently, leads to lower surface concentrations.

It is also possible that we might have underestimated the EC emissions from the brick kilns near Bode. In our emission inventory we use an average kiln productivity to estimate the emissions, thus if the productivities of these kilns are higher than the average, or if their efficiency is lower, their emissions could have been underestimated in the inventory. Another possibility is that additional sources are still missing, for example, garbage burning, biomass burning, and/or diesel generators. Stockwell et al. (2016) found that garbage burning in the Kathmandu Valley may produce significantly more EC emissions than previously thought. Due to power outages, especially in the dry season, the use of generators was still prevalent in the valley in 2015. A study conducted by the World Bank (2014) found that nearly 200,000 small power generators, powered by diesel, were used for pervasive power shortages in Nepal. Nevertheless, these comparisons suggest that there is still a need for further improvement of constructing local emissions inventories in the Kathmandu Valley.

## 4.4 SO$_2$

Fig. 10 presents the two-week average SO$_2$ concentrations for the three simulations. The average SO$_2$ concentration in the HTAP_vehicle_brick simulation is approximately 3.6 $\mu$g m$^{-3}$ in the Kathmandu Valley, higher than the surrounding areas in Nepal. Revised vehicle emissions have little impact on SO$_2$ concentration in the valley, but brick kiln emissions contribute 50% of simulated SO$_2$ concentrations. The two-week mean SO$_2$ concentration measured at the Bode site was 39.7 $\mu$g m$^{-3}$. The model largely underestimates the observation, with the modeled mean SO$_2$ concentration of 5.3 $\mu$g m$^{-3}$. SO$_2$ is mainly a primary pollutant, directly emitted from sources. SO$_2$ concentrations are highly related to its emissions. The large discrepancy between the model simulation and observation at this site is probably because our model resolution is not able to capture spatially-concentrated high emissions of SO$_2$ near Bode. Kiros et al. (2016) measured SO$_2$ concentrations at 15 sites in 8 weeks from March to May 2013 in the Kathmandu Valley and observed high SO$_2$ concentrations close to brick kilns. Particularly, the average SO$_2$ concentration measured at Bode was the highest (39.2 $\mu$g m$^{-3}$) and 2-6 times higher than those at other 14 sites (4.7-15.8 $\mu$g m$^{-3}$) in the Kathmandu Valley. They explained that the elevated surface SO$_2$ concentration at Bode is mainly caused by nearby brick kilns, which are fueled by coal. We found that there are 12 brick kilns located within the 4 km distance from the Bode site. Although we have included emissions of these brick kilns in our model inputs, we may still have over-estimated dilution and/or underestimated emissions from these brick kilns by using monthly average productivity and average emission factors of zigzag brick kilns. We hope to improve our emission inventory of brick kilns when more information of individual brick kilns becomes available.

Fig. 12 compares modeled two-week mean SO$_2$ concentrations with observed eight-week mean SO$_2$ measured between March 23 and May 18, 2013, reported in the study of Kiros et al. (2016). None of these sites exceeded the Nepal national air quality standard of 70 $\mu$g m$^{-3}$ for 24 h mean, but SO$_2$ concentrations at Bode site were almost twice as high as the WHO standard of 20 $\mu$g m$^{-3}$. Since our own NAMaSTE campaign only collected SO$_2$ at the Bode site, we also included the study of Kiros et al. (2016) to illustrate the magnitude difference in observational data at different locations within the Kathmandu Valley. The two-week mean SO$_2$ concentration from the NAMaSTE in 2015 was 39.7 $\mu$g m$^{-3}$ at Bode, while the 8-week mean in 2013 by Kiros et al. (2016) was 39.2 $\mu$g m$^{-3}$, showing similarities, giving us confidence that comparing the magnitude difference among sites was possible, despite the difference in observed years. We used these 2013 measurements to represent the ambient SO$_2$ concentrations during our simulation period. Including brick kilns emissions improves model prediction of SO$_2$ concentrations at all sites. The simulated SO$_2$ concentrations with brick kiln emissions are closer to observations compared with those without them, although the model still underestimates SO$_2$. This underestimation is probably due to brick kiln SO$_2$ emissions. We applied an emission factor of 12.7 g/kg of fuel measured from zigzag kilns (Stockwell et al., 2016) to all types of brick kilns. This was the only available observational data in Nepal at the time of this study. A more recent study by Nepal et al. (2019) reported that the mean value of SO$_2$ emission factor from zigzag kilns is 24 $\pm$ 22 g/kg fuel, which is almost twice as high as that used in our study. If we doubled our SO$_2$ emissions for brick kilns, the modeled SO$_2$ concentrations would be much closer to the observations. Assuming the linear relationship in SO$_2$, the average difference between the observed and modeled SO$_2$ concentrations would drop from 4.4 $\mu$g m$^{-3}$ to 2.8 $\mu$g m$^{-3}$. We plan to revisit our brick kiln emissions inventory,

as more emission factors become available. Our study highlights the importance of improving emission factor of $SO_2$ for brick kilns in Nepal. The difference between observation and model simulation ranges from 0.8 to 10 $\mu$g m$^{-3}$ for all sites except Bode where the difference is 34.4 $\mu$g m$^{-3}$. This result suggests that surface $SO_2$ concentrations in the Kathmandu Valley are highly variable and are influenced by nearby sources. Future simulations with a higher spatial model resolution and emissions
inputs may help to resolve the strong spatial gradients in $SO_2$ concentrations.

## 5   Summary and future work

In this paper, we modified the HTAP emissions inventory for Kathmandu's road transport sector. We also developed a point source emissions inventory for brick kilns in the Kathmandu Valley, and examined the impacts of emissions from on-road vehicles and brick kilns on local air quality for April 2015. Emissions from vehicles were updated using the IVE model to reflect
the most recent vehicle registration information and the local vehicle technology and driving conditions in the Kathmandu Valley. We found that PM emissions from the road transport sector in the HTAP_v2.2 inventory are largely underestimated. The IVE-estimated EC emissions are 375 times higher than those estimated in HTAP. Our brick kiln emissions estimates were created to account for one of the most important missing sources in the existing emission inventories. We found that emissions from brick kilns contribute 68% of the total $SO_2$ emissions in the Kathmandu Valley in HTAP_v2.2 inventory.

Using the original HTAP emissions results in large underestimations of both surface EC and $SO_2$ in the Kathmandu Valley. Our revised vehicle emissions significantly reduced model bias and improved model-observation correlation for surface EC concentrations. We found that surface EC concentrations increased by 50% on average due to our revised on-road vehicle emissions estimates. On the other hand, brick kiln emissions contributed approximately 50% of the modeled surface $SO_2$ concentrations in the Kathmandu Valley. Although model performance has been enhanced considerably, by using revised
vehicle emissions and by adding newly-created brick kiln emissions, the model still underestimates the observed EC by 73% and $SO_2$ by 87% at the Bode site during the simulation period. The large underestimation at Bode could be a result of the site's proximity to large point sources or assuming average EFs for these point sources, but additional sources not included in our inventory could also be important for improving the model performance. More information on the production rates of individual brick kilns and emission factors for each major type of brick kiln could significantly improve the inventory and
comparisons. It is important that the complex topography and meteorology with limited observational data limits the degree of model evaluation currently possible in the Kathmandu Valley. The concentrations of pollutants are highly dependent on the measurement locations and topography of their surroundings and more observational data at a finer scale within the valley is essential to better evaluate the local chemical transport models. Despite the uncertainties, the results here suggest that emissions from brick kilns are substantial and current estimates of emissions are underestimating total emissions by omitting this source.
Our main objective was to improve the emission inventories for on-road vehicles and brick kilns and assess the impacts of revised emissions on local air quality. Our results demonstrate that the existing emissions inventories need significant modification for the road transportation sector. Missing sources in the Kathmandu Valley such as brick kilns are also important in predicting local air quality. We suggest that more efforts are needed to improve local emissions information by updating emis-

sions estimates from major sources and developing an emissions inventory including underrepresented sources, such as crop residue and garbage burning. A more comprehensive and accurate emission inventory allows the local government to identify and define key emission sources in the Kathmandu Valley. The improved emission inventory is urgently needed to robustly evaluate the effectiveness of various future policies on emission mitigation in this region.

*Code availability:* The WRF-Chem model is an open-source, publicly available, and continually improved software. The version 3.5 used in this study can be downloaded at http://www2.mmm.ucar.edu/wrf/users/download/get_source.html. Known problems of the WRF-Chem version 3.5 have been fixed, using solutions provided online at http://ruc.noaa.gov/wrf/WG11/known-prob_v3.5.htm. The revised gridded vehicle and brick kiln emissions for the Kathmandu Valley will be available in PANGAEA.

*Author contribution:* MZ ran the model simulations and drafted the paper. ES and AA contributed to the analysis and the writing of all versions of the paper. CC and BS contributed to developing new brick kiln emissions inventory for Nepal. WY contributed to transport emissions analysis. WC, RJY, TJ, EAS, MR, and AKP provided observational data and support throughout the manuscript production. All authors contributed to revising the paper.

*Competing interests:* The authors declare that they have no conflict of interest.

*Acknowledgements.* The NCEP GFS data used for this study are from the Research Data Archive (RDA) which is maintained by the Computational and Information Systems Laboratory (CISL) at the National Center for Atmospheric Research (NCAR). The data are available at http://rda.ucar.edu/datasets/ds083.0/. This study was supported by the National Science Foundation (Grant numbers AGS-1350021, AGS-1349976, AGS-1351616, and AGS-1355551). Additional support was provided by ICIMOD through a contract with the University of Virginia.

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

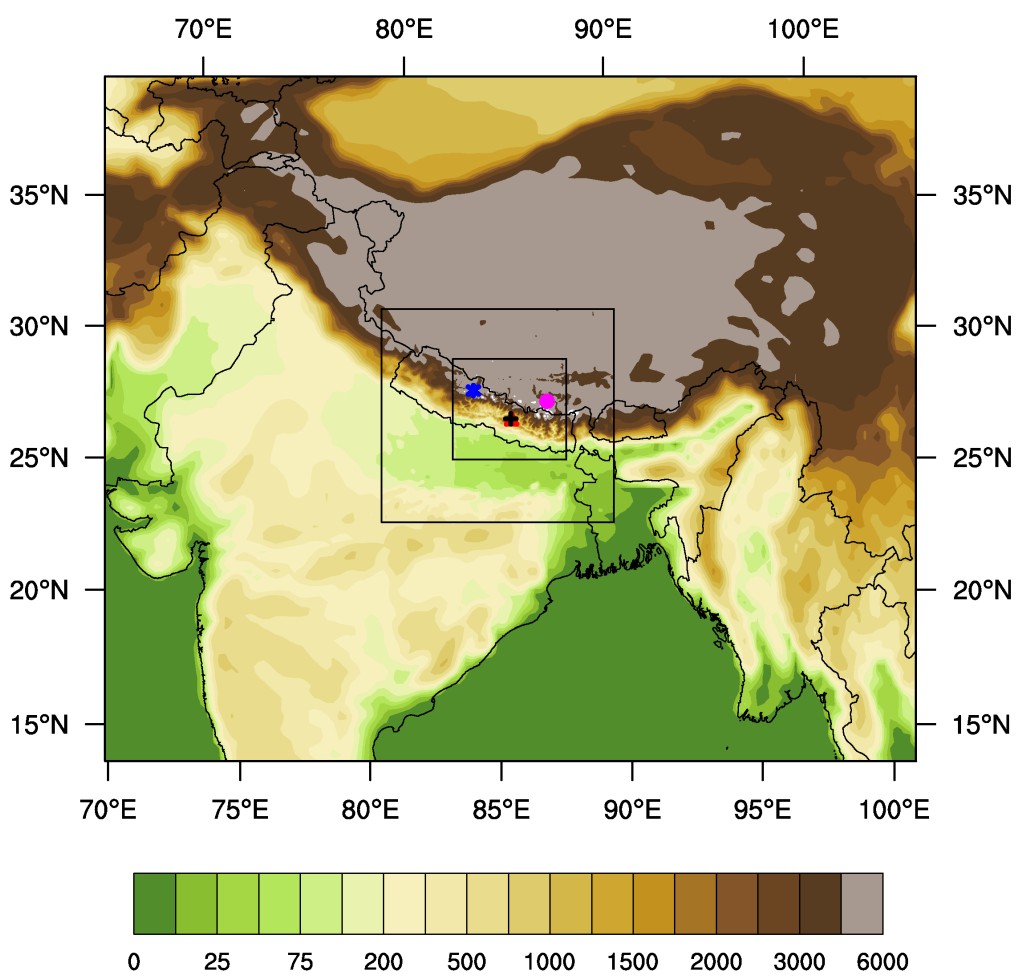

**Figure 1.** Nested model domains with the terrain heights in meters (color shaded) and the locations of four measurement sites: Jomsom, Nepal (blue asterisk); QOMS_CAS, China (pink dot); Bode, Nepal (red triangle); and the Tribhuvan International Airport, Nepal (black cross).

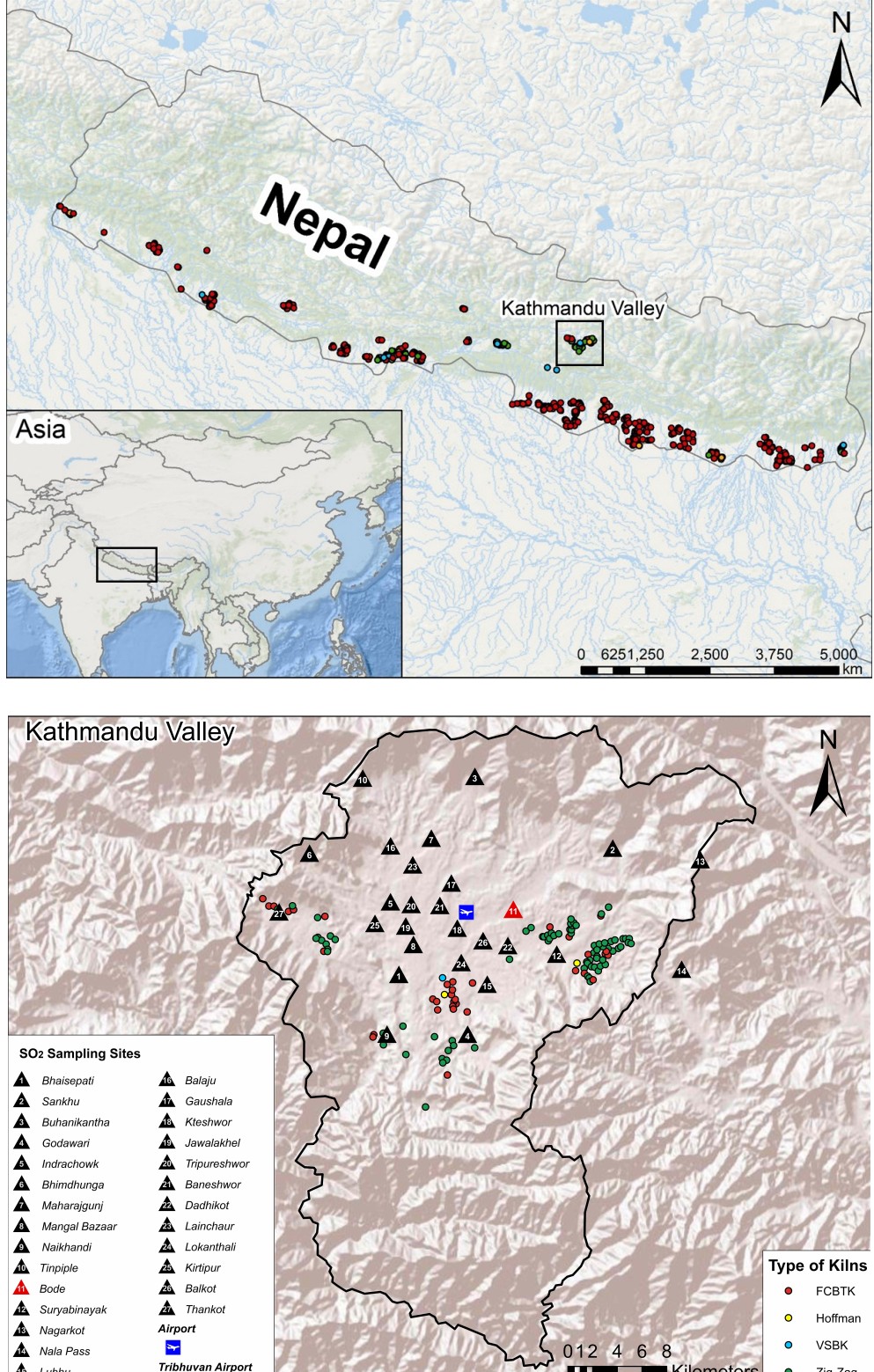

**Figure 2.** Spatial distribution of brick kilns in: Nepal (top) and Kathmandu Valley (bottom). Red, orange, blue, and green dotes denote the Fixed Chimney Bull's Trench Kiln (FCBTK), Hoffman kiln, Vertical Shaft Brick Kiln (VSBK), and Zigzag kiln, respectively.

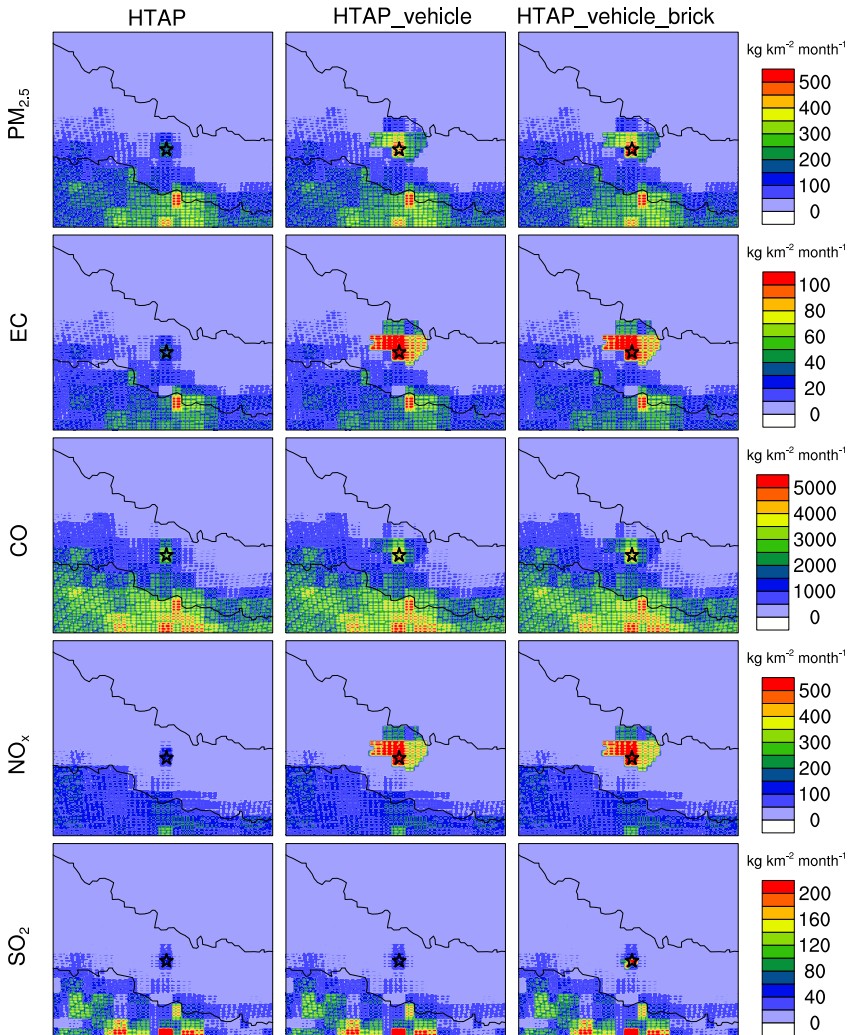

**Figure 3.** The monthly mean surface emissions of five pollutants, $PM_{2.5}$, EC, CO, $NO_x$, and $SO_2$ from all sources in April 2015 used in WRF-Chem for the three simulations. The star indicates the location of the Bode site.

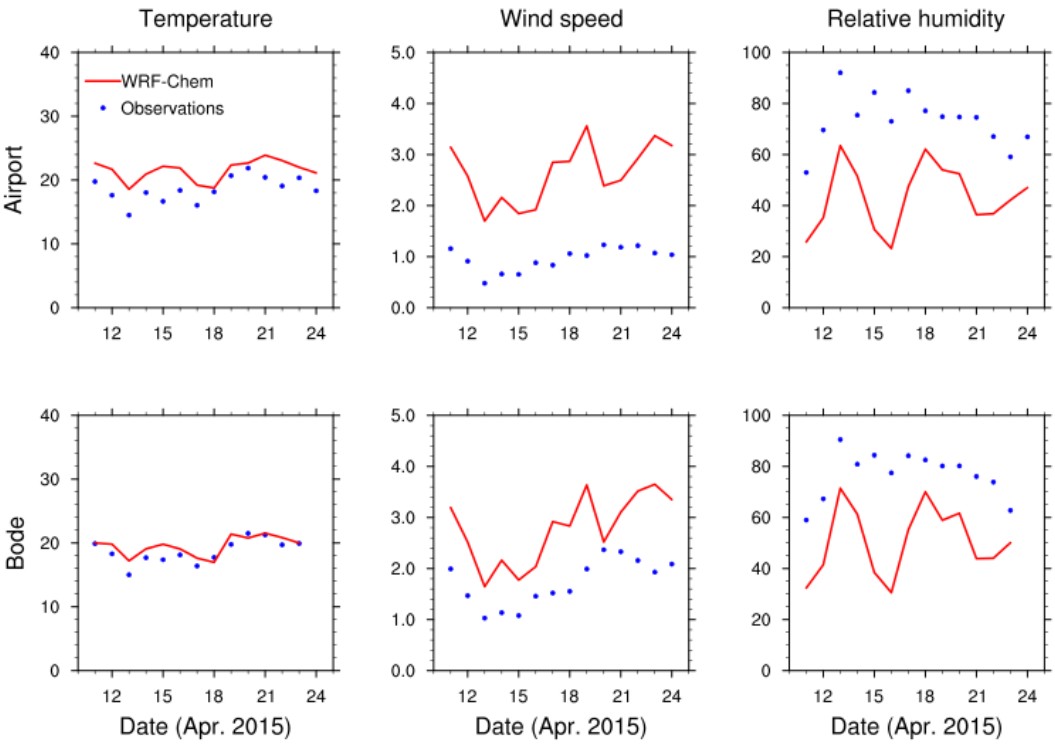

**Figure 4.** Comparisons of observed (blue dots) and modeled (red lines) daily mean 2-m temperature, 10-m wind speed, and 2-m relative humidity at two sites (Airport and Bode) in the Kathmandu Valley.

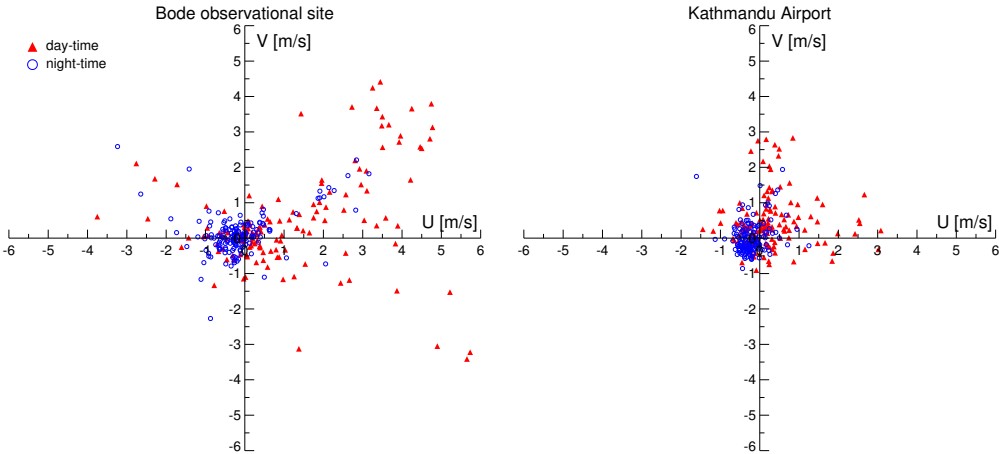

**Figure 5.** Hourly-averaged observed U- and V-wind components during the period of April 11-24 at: (a) Bode site and (b) Tribhuvan International Airport. Daytime winds are shown with red closed triangles and the open blue circles denote winds during the night. Note that the winds at Bode site and the airport are measured at a height of 17 and 10 m above surface, respectively.

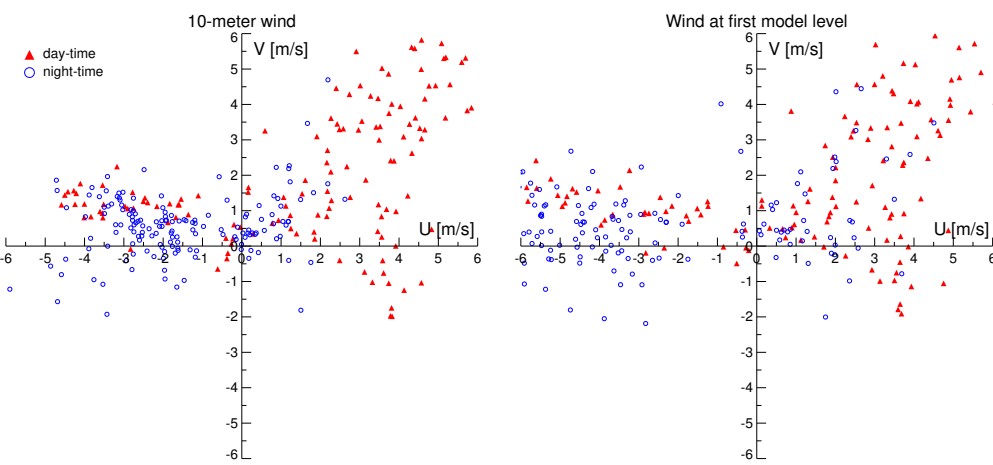

**Figure 6.** Hourly-averaged simulated U- and V-wind components during the period of April 11-24: (a) at 10-m height above surface and (b) at the first model level ( 28 m height) at the model grid point closest to Bode site. Daytime winds are shown with red closed triangles and the open blue circles denote winds during the night.

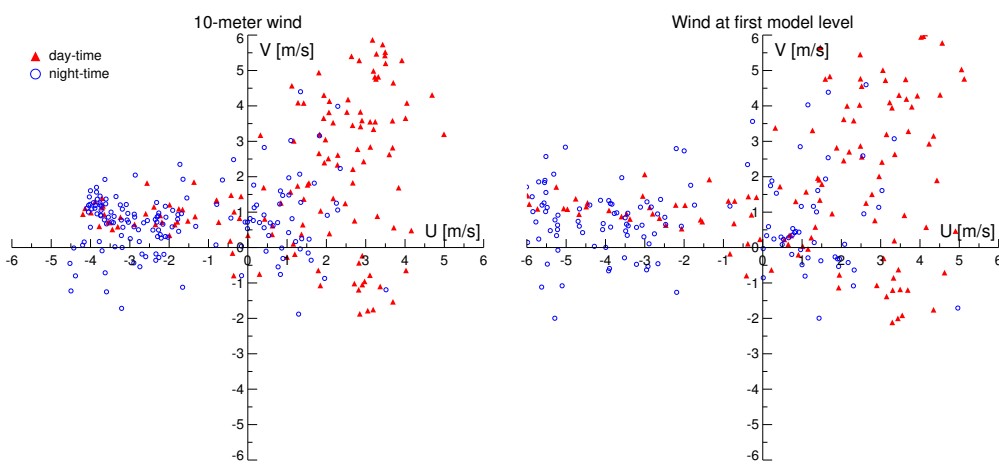

**Figure 7.** Same as Fig. 6 but for the model grid point closest to the Tribhuvan International Airport.

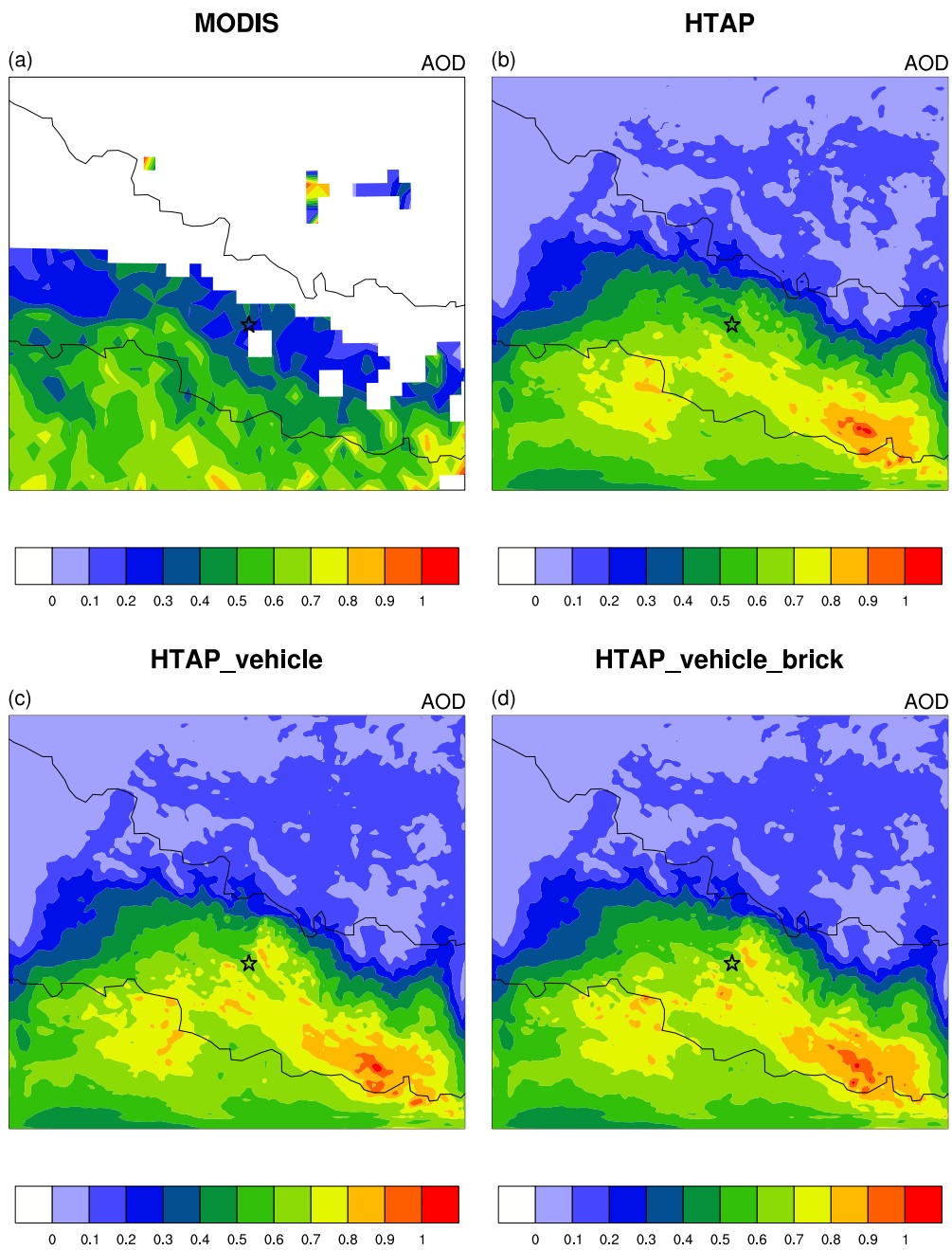

**Figure 8.** Two-week average AOD: (a) retrieved from MODIS/Terra satellite; (b) simulated using WRF-Chem with original HTAP emissions; (c) simulated using WRF-Chem with HTAP with updated vehicle emissions, and (d) simulated using WRF-Chem with HTAP emissions plus updated vehicles and brick kilns emissions. The star indicates the location of the Bode site.

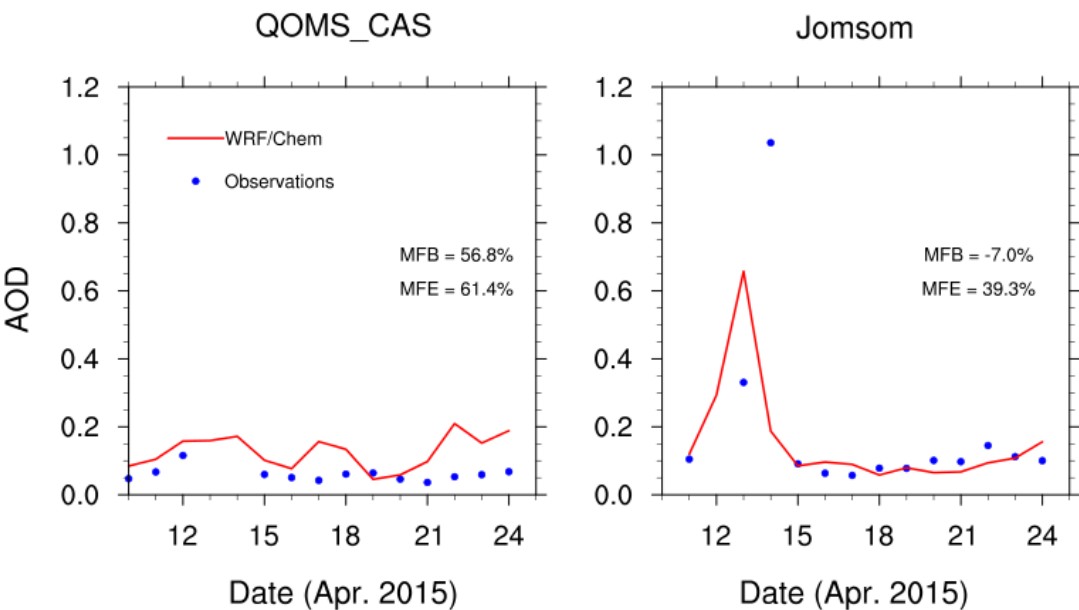

**Figure 9.** Comparisons of observed (blue dots) and modeled (red lines) daily mean AOD at two AERONET sites (QOMS_CAS and Jomsom).

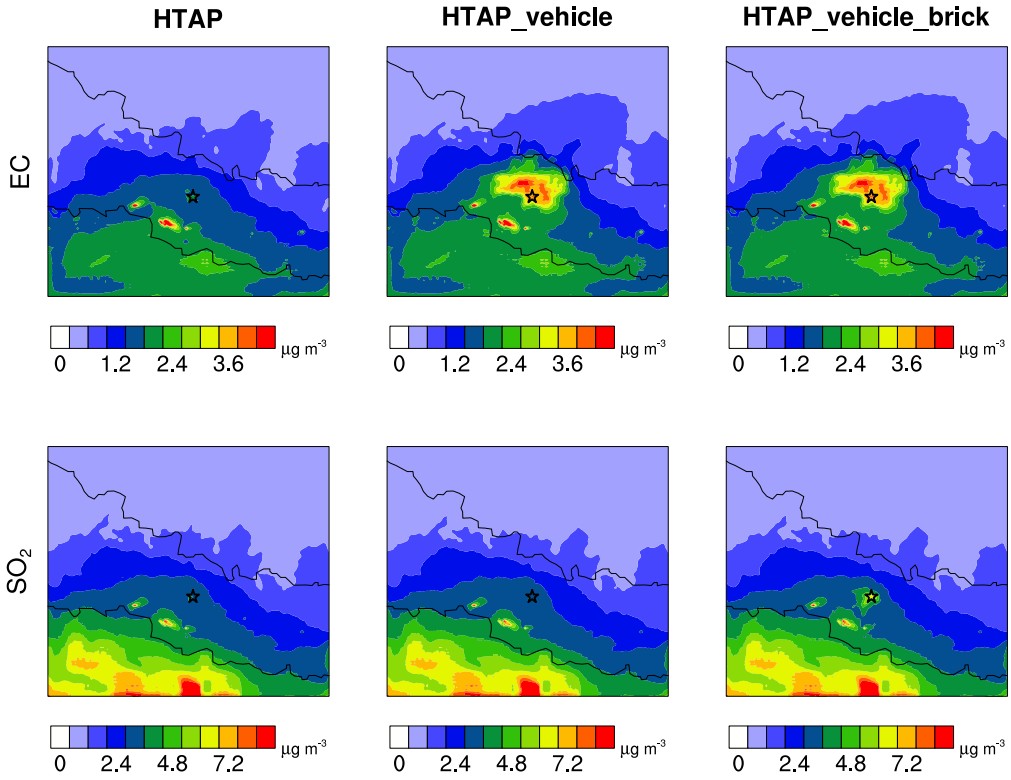

**Figure 10.** Two-week average surface EC (top) and SO₂ (bottom) concentrations obtained from three simulations: HTAP, HTAP_vehicle, and HTAP_vehicle_brick. The star indicates the location of the Bode site.

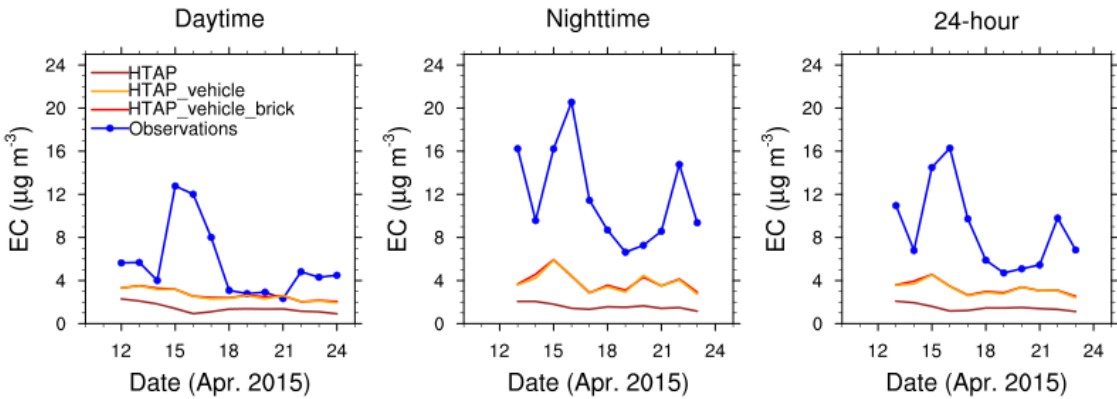

**Figure 11.** Comparisons of observed (blue dots) and modeled EC concentrations in daytime, nighttime, and daily mean for the three scenarios at Bode. Observed values are taken during the NAMaSTE campaign (Jayarathne et al., 2018)

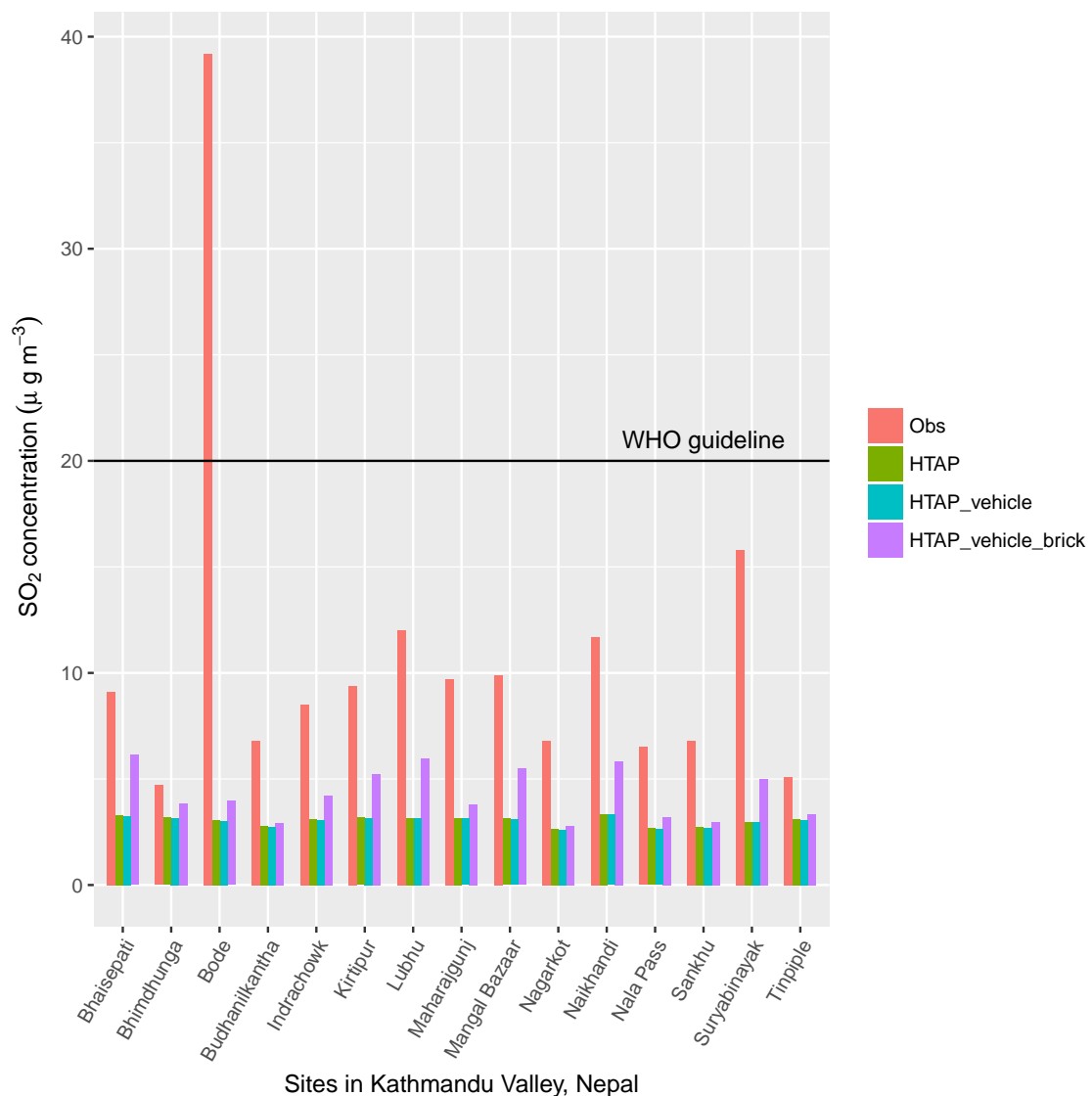

**Figure 12.** Comparisons of modeled and observed SO$_2$ concentrations at 14 sites in the Kathmandu Valley. The modeled SO$_2$ is the 2-week mean daily SO$_2$ concentrations averaged from April 12-24, 2015. The observed SO2 is the 8-week mean SO$_2$ concentrations between March 23 and May 18, 2013 reported in the study of Kiros et al. (2016).

**Table 1.** Description for each emissions scenarios in 2015

| Emission Scenario | Description |
|---|---|
| HTAP | Original HTAP_v2.2 emission inventory |
| HTAP_vehicle | Original HTAP_v2.2 + updated vehicle emissions |
| HTAP_vehicle_brick | Original HTAP_v2.2 + updated vehicle emissions + brick kiln emissions |

**Table 2.** Number of vehicles, daily mileage and starts for vehicles in the Bagmati Zone, Nepal in 2015.

| Vehicle category | Number of vehicles, 2015[a] | daily VKT (km day$^{-1}$)[b] | Number of starts per day |
|---|---|---|---|
| Motorcycle | 722695 | 15 | 3.8 |
| Bus/mini bus | 20207 | 96 | 9 |
| Taxi | 6206 | 87 | 15 |
| Car/pickup/jeep | 136391 | 44 | 15 |
| Van/microbus | 2123 | 42 | 10.3 |
| 3-wheeler (tempo) | 2528 | 63 | 12 |
| Truck/mini truck | 18917 | 107 | 9 |

a: We obtained the number of vehicles in 2015 from the Department of Transport, Nepal; b: VKT of Truck/mini truck are from Malla et al. (2014)

**Table 3.** Total emissions from vehicles, brick kilns in the Kathmandu Valley during April 2015 estimated by this study versus corresponding emissions from vehicles and all sources considered in HTAP emissions inventory.

| Unit (ton month$^{-1}$) | CO | SO$_2$ | NO$_x$ | NMVOCs | EC | OC | PM$_{2.5}$ |
|---|---|---|---|---|---|---|---|
| Total vehicle emissions, this study | 6551 | 41 | 6152 | 1413 | 827 | 234 | 1852 |
| Total brick kilns emission, this study | 98 | 123 | 13 | 13 | 1.08 | 10 | 135 |
| Total emissions from all sectors, this study | 18668 | 308 | 6565 | 3978 | 976 | 839 | 2796 |
| Total transport sector emissions in HTAP, 2010 | 188 | 35 | 80 | 98 | 2.2 | 2.3 | 10 |
| Total emissions from all sectors in HTAP, 2010 | 12207 | 179 | 479 | 2651 | 150 | 598 | 819 |

**Table 4.** Statistical performance of model simulation for daily surface temperature, 10-m wind speed, and surface relative humidity at the Tribhuvan International Airport (Airport) and Bode

| Statistical metrics | | Surface Temperature (°C) | | 10-m Wind Speed (m s$^{-1}$) | | Surface RH (%) | |
|---|---|---|---|---|---|---|---|
| | | Airport | Bode | Airport | Bode | Airport | Bode |
| Mean | Observation | 18.6 | 18.7 | 1.0 | 1.7 | 73.3 | 76.9 |
| | Modeled | 21.5 | 19.6 | 2.6 | 2.8 | 43.5 | 50.7 |
| Min/Max | Observation | 14.5/21.9 | 15.0/21.5 | 0.5/1.2 | 1.0/2.4 | 53.0/92.1 | 59.0/90.5 |
| | Modeled | 18.6/23.9 | 17.0/21.6 | 1.7/3.6 | 1.7/3.7 | 23.2/63.5 | 30.6/71.5 |
| Mean bias | | 2.9 | 0.9 | 1.7 | 1.1 | -29.9 | -26.2 |
| NMB (%) | | 15.8 | 4.7 | 176.0 | 61.2 | -40.8 | -34.0 |
| RMSE | | 3.2 | 1.3 | 1.7 | 1.1 | 32.0 | 28.2 |
| Correlation | | 0.7 | 0.8 | 0.7 | 0.8 | 0.5 | 0.6 |

**Table 5.** Statistical measures calculated for three model simulations with different emissions inputs for EC. Obs ($\mu$g m$^{-3}$) and Model ($\mu$g m$^{-3}$) are 2-week mean daily average value of observed and modeled EC, respectively. $r$ is correlation coefficient between observation and model simulations; NMB (%) is the normalized mean bias between observations and model simulations; MFB (%) and MFE (%) are the mean fractional bias and mean fractional error; RMSE is the root mean square error between observations and model ($\mu$g m$^{-3}$).

| Emissions | Day/Night | Obs | Model | $r$ | MB | NMB | MFB | MFE | RMSE |
|---|---|---|---|---|---|---|---|---|---|
| | Day | 5.60 | 1.34 | -0.21 | -4.27 | -76.15 | -107.71 | 107.71 | 5.49 |
| HTAP | Night | 10.82 | 1.61 | 0.12 | -9.20 | -85.08 | -142.45 | 142.45 | 10.25 |
| | 24-h | 8.32 | 1.48 | 0.19 | -6.83 | -82.19 | -125.77 | 125.77 | 8.31 |
| | Day | 5.60 | 2.56 | 0.28 | -3.04 | -54.32 | -58.74 | 60.62 | 4.47 |
| HTAP_vehicle | Night | 10.82 | 3.77 | 0.48 | -7.05 | -65.17 | -90.56 | 90.56 | 8.20 |
| | 24-h | 8.32 | 3.19 | 0.61 | -5.13 | -61.66 | -75.29 | 76.19 | 6.67 |
| | Day | 5.60 | 2.62 | 0.25 | -2.99 | -53.31 | -56.68 | 58.52 | 4.44 |
| HTAP_vehicle_brick | Night | 10.82 | 3.85 | 0.47 | -6.97 | -64.45 | -88.69 | 88.69 | 8.13 |
| | 24-h | 8.32 | 3.26 | 0.61 | -5.06 | -60.85 | -73.33 | 74.21 | 6.62 |

**Appendix A: Supplementary Materials**

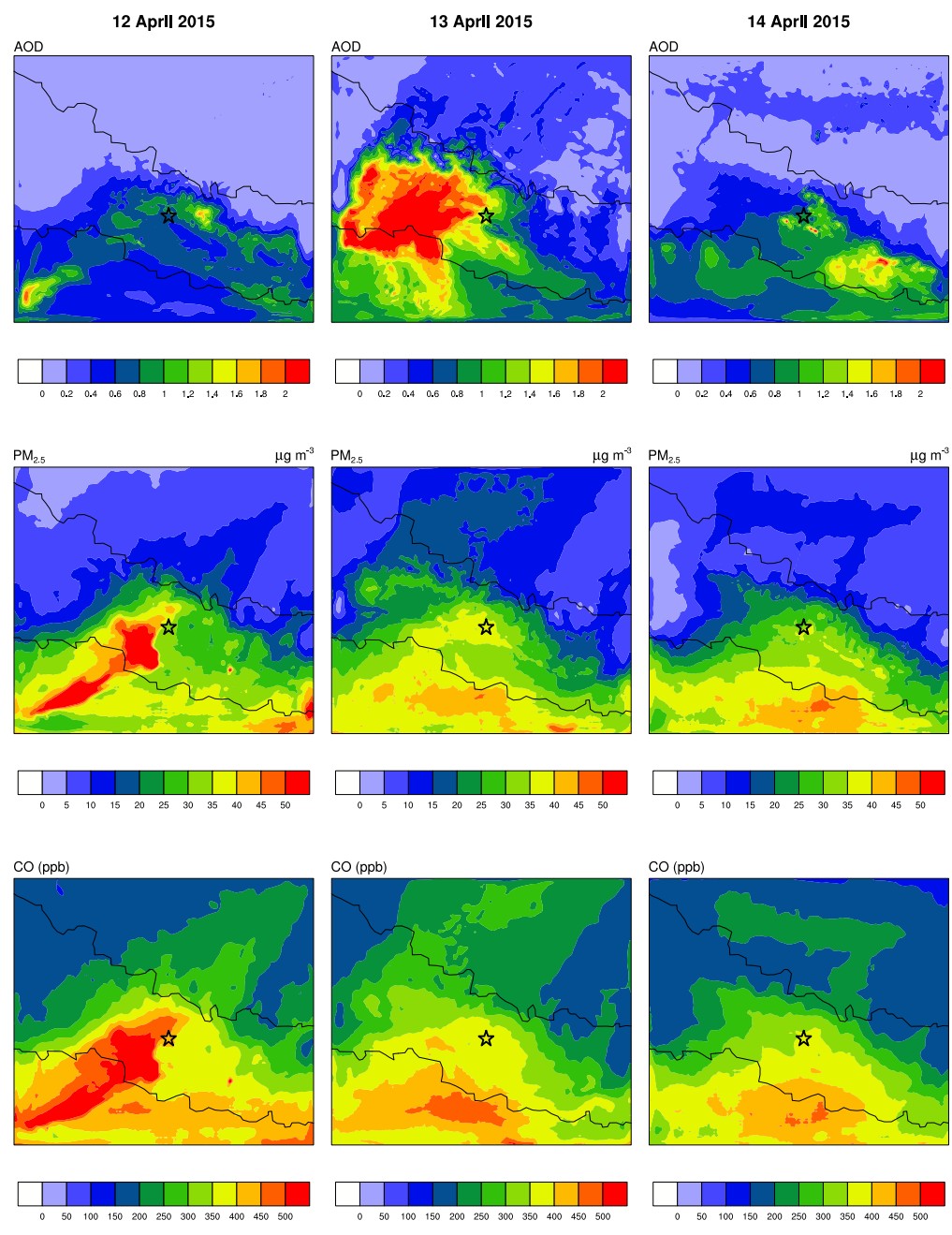

**Figure S-11.** Simulated daily AOD, surface PM$_{2.5}$ and CO in April 12-14, 2015. Kathmandu Valley is indicated with an open star in the figure.

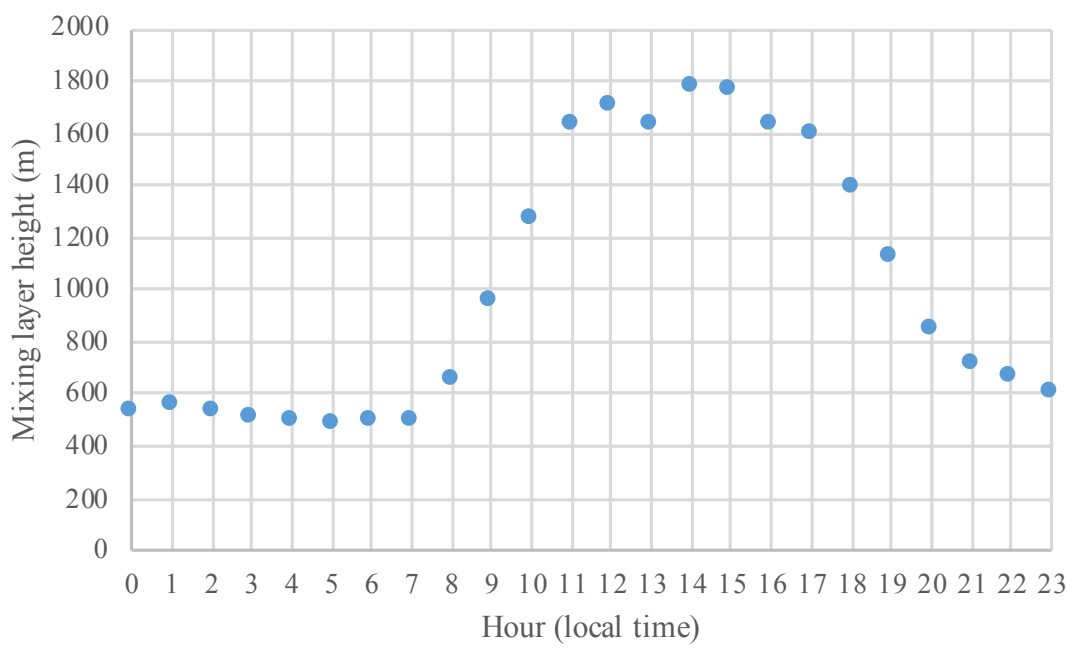

**Figure S-10.** Averaged diurnal cycle of the mixing layer height (m) as simulated by WRF-Chem for the period April 12-24, 2015.

**Table S-4a.** Vehicle technology used as IVE model inputs in 2015

| Description | Fuel | Weight | Air/Fuel Control | Exhaust Control | Evaporative Control | Age (km travelled) | Index* | Share | Corresponding Euro Standards |
|---|---|---|---|---|---|---|---|---|---|
| | | | | Motorcycle | | | | | |
| Small Engine | Petrol | Light | 4-Cycle, Carb | None | None | >50K | 1208 | 0.010 | Pre-Euro |
| Small Engine | Petrol | Light | 4-Cycle, Carb | Improved | None | 26-50K | 1216 | 0.010 | Pre-Euro |
| Small Engine | Petrol | Light | 4-Cycle, Carb | Improved | None | >50K | 1217 | 0.040 | Pre-Euro |
| Small Engine | Petrol | Medium | 4-Cycle, Carb | Improved | None | 26-50K | 1219 | 0.010 | Pre-Euro |
| Small Engine | Petrol | Medium | 4-Cycle, Carb | Improved | None | >50K | 1220 | 0.020 | Pre-Euro |
| Small Engine | Petrol | Light | 4-Cycle, Carb | High Tech | None | 0-25K | 1224 | 0.010 | Euro III |
| Small Engine | Petrol | Light | 4-Cycle, Carb | High Tech | None | 26-50K | 1225 | 0.040 | Pre-Euro |
| Small Engine | Petrol | Light | 4-Cycle, Carb | High Tech | None | >50K | 1226 | 0.020 | Pre-Euro |
| Small Engine | Petrol | Medium | 4-Cycle, Carb | High Tech | None | 0-25K | 1227 | 0.030 | Euro III |
| Small Engine | Petrol | Medium | 4-Cycle, Carb | High Tech | None | 26-50K | 1228 | 0.030 | Euro III |
| Small Engine | Petrol | Medium | 4-Cycle, Carb | High Tech | None | >50K | 1229 | 0.040 | Pre-Euro |
| Small Engine | Petrol | Light | 4-Cycle, Carb | Catalyst | None | 0-25K | 1233 | 0.020 | Euro III |
| Small Engine | Petrol | Light | 4-Cycle, Carb | Catalyst | None | 26-50K | 1234 | 0.030 | Euro III |
| Small Engine | Petrol | Light | 4-Cycle, Carb | Catalyst | None | >50K | 1235 | 0.010 | Pre-Euro |
| Small Engine | Petrol | Medium | 4-Cycle, Carb | Catalyst | None | 0-25K | 1236 | 0.450 | Euro III |
| Small Engine | Petrol | Medium | 4-Cycle, Carb | Catalyst | None | 26-50K | 1237 | 0.110 | Euro III |
| Small Engine | Petrol | Medium | 4-Cycle, Carb | Catalyst | None | >50K | 1238 | 0.040 | Pre-Euro |
| Small Engine | Petrol | Medium | 4-Cycle, FI | Catalyst | PCV | 0-25K | 1245 | 0.070 | Euro III |
| Small Engine | Petrol | Medium | 4-Cycle, FI | Catalyst | PCV | 26-50K | 1246 | 0.010 | Pre-Euro |
| | | | | Bus/Mini bus | | | | | |
| Truck/Bus | Diesel | Heavy | Pre-Chamber Injection | None | None | >161K km | 1079 | 0.100 | Pre-Euro |
| Truck/Bus | Diesel | Heavy | Direct Injection | EGR+Improved | None | 80-161K km | 1096 | 0.010 | Pre-Euro |
| Truck/Bus | Diesel | Heavy | Direct Injection | EGR+Improved | None | >161K km | 1097 | 0.010 | Pre-Euro |
| Truck/Bus | Diesel | Heavy | FI | Particulate/NOx | None | 80-161K km | 1114 | 0.010 | Pre-Euro |
| Truck/Bus | Diesel | Heavy | FI | Particulate/NOx | None | >161K km | 1115 | 0.050 | Pre-Euro |
| Truck/Bus | Diesel | Heavy | FI | EuroI | None | <79K km | 1122 | 0.010 | Euro I |
| Truck/Bus | Diesel | Heavy | FI | EuroI | None | 80-161K km | 1123 | 0.020 | Euro I |
| Truck/Bus | Diesel | Heavy | FI | EuroI | None | >161K km | 1124 | 0.190 | Euro I |
| Truck/Bus | Diesel | Heavy | FI | EuroII | None | <79K km | 1131 | 0.320 | Euro II |
| Truck/Bus | Diesel | Heavy | FI | EuroII | None | 80-161K km | 1132 | 0.100 | Euro II |
| Truck/Bus | Diesel | Heavy | FI | EuroII | None | >161K km | 1133 | 0.180 | Euro II |
| | | | | Taxi | | | | | |
| Auto/Small Truck | Petrol | Light | Multi-Pt FI | none | PCV | 80-161K km | 100 | 0.03 | Pre-Euro |
| Auto/Small Truck | Petrol | Light | Multi-Pt FI | none | PCV | >161K km | 101 | 0.28 | Pre-Euro |
| Auto/Small Truck | Petrol | Light | Multi-Pt FI | EuroI | PCV/Tank | <79K km | 171 | 0.04 | Euro I |
| Auto/Small Truck | Petrol | Light | Multi-Pt FI | EuroI | PCV/Tank | 80-161K km | 172 | 0.11 | Euro I |
| Auto/Small Truck | Petrol | Light | Multi-Pt FI | EuroI | PCV/Tank | >161K km | 173 | 0.45 | Euro I |
| Auto/Small Truck | Petrol | Light | Multi-Pt FI | EuroII | PCV/Tank | <79K km | 180 | 0.04 | Euro II |
| Auto/Small Truck | Petrol | Light | Multi-Pt FI | EuroII | PCV/Tank | 80-161K km | 181 | 0.02 | Euro II |
| Auto/Small Truck | Petrol | Light | Multi-Pt FI | EuroII | PCV/Tank | >161K km | 182 | 0.03 | Euro II |

*: Index is a serial of number from 0 to 1371 label the 1371 types of vehicle technologies that used in the IVE model.

**Table S-4b.** Vehicle technology used as IVE model inputs in 2015

| | | | | | | | | | |
|---|---|---|---|---|---|---|---|---|---|
| Car/Pickup | | | | | | | | | |
| Auto/Small Truck | Petrol | Medium | Carburetor | 2-Way | PCV | <79K km | 12 | 0.004 | Euro I |
| Auto/Small Truck | Petrol | Light | Carburetor | 3-Way | PCV | <79K km | 27 | 0.006 | Euro I |
| Auto/Small Truck | Petrol | Medium | Carburetor | 3-Way | PCV | <79K km | 30 | 0.003 | Euro I |
| Auto/Small Truck | Petrol | Light | Single-Pt FI | 2-Way | PCV | <79K km | 63 | 0.325 | Euro I |
| Auto/Small Truck | Petrol | Light | Single-Pt FI | 2-Way | PCV | 80-161K km | 64 | 0.013 | Euro I |
| Auto/Small Truck | Petrol | Medium | Single-Pt FI | 2-Way | PCV | <79K km | 66 | 0.029 | Euro I |
| Auto/Small Truck | Petrol | Light | Multi-Pt FI | 3-Way | PCV | <79K km | 117 | 0.120 | Euro II |
| Auto/Small Truck | Petrol | Medium | Multi-Pt FI | 3-Way | PCV | <79K km | 120 | 0.052 | Euro II |
| Auto/Small Truck | Propane | Light | Carb/Mixer | None | PCV | <79K km | 396 | 0.006 | Pre-Euro |
| Auto/Small Truck | Propane | Light | Carb/Mixer | None | PCV | 80-161K km | 397 | 0.003 | Pre-Euro |
| Auto/Small Truck | Propane | Light | Carb/Mixer | 3-Way | PCV | <79K km | 423 | 0.003 | Euro I |
| Auto/Small Truck | Diesel | Light | Pre-Chamber Inject. | Improved | None | <79K km | 747 | 0.198 | Euro I |
| Auto/Small Truck | Diesel | Light | Pre-Chamber Inject. | Improved | None | 80-161K km | 748 | 0.016 | Euro I |
| Auto/Small Truck | Diesel | Light | Pre-Chamber Inject. | Improved | None | >161K km | 749 | 0.003 | Euro I |
| Auto/Small Truck | Diesel | Medium | Pre-Chamber Inject. | Improved | None | <79K km | 750 | 0.094 | Euro I |
| Auto/Small Truck | Diesel | Medium | Pre-Chamber Inject. | Improved | None | 80-161K km | 751 | 0.071 | Euro I |
| Auto/Small Truck | Diesel | Medium | Pre-Chamber Inject. | Improved | None | >161K km | 752 | 0.026 | Euro I |
| Auto/Small Truck | Diesel | Heavy | Pre-Chamber Inject. | Improved | None | <79K km | 753 | 0.023 | Euro I |
| Auto/Small Truck | Diesel | Heavy | Pre-Chamber Inject. | Improved | None | >161K km | 755 | 0.003 | Euro I |
| Van/Jeep | | | | | | | | | |
| Auto/Small Truck | Diesel | Medium | Direct Injection | EGR+Improved | None | >161K km | 761 | 0.02 | Pre-Euro |
| Auto/Small Truck | Diesel | Medium | FI | EuroI | None | <79K km | 786 | 0.02 | Euro I |
| Auto/Small Truck | Diesel | Medium | FI | EuroI | None | 80-161K km | 787 | 0.01 | Euro I |
| Auto/Small Truck | Diesel | Medium | FI | EuroI | None | >161K km | 78 | 0.92 | Euro I |
| Auto/Small Truck | Diesel | Heavy | FI | EuroI | None | <79K km | 789 | 0.03 | Euro I |
| 3-Wheeler | | | | | | | | | |
| Small Engine | CNG/LPG | Heavy | 4-Cycle, Carb | Catalyst | None | 26-50K | 1276 | 0.02 | Pre-Euro |
| Small Engine | CNG/LPG | Heavy | 4-Cycle, Carb | Catalyst | None | >50K | 1277 | 0.98 | Pre-Euro |
| Truck/Mini truck | | | | | | | | | |
| Truck/Bus | Diesel | Light | Pre-Chamber Inject. | None | None | >161K km | 1073 | 0.09605 | Pre-Euro |
| Truck/Bus | Diesel | Medium | Pre-Chamber Inject. | None | None | >161K km | 1076 | 0.08145 | Pre-Euro |
| Truck/Bus | Diesel | Heavy | Pre-Chamber Inject. | None | None | >161K km | 1079 | 0.1775 | Pre-Euro |
| Truck/Bus | Diesel | Light | Direct Injection | Improved | None | >161K km | 1082 | 0.10275 | Pre-Euro |
| Truck/Bus | Diesel | Medium | Direct Injection | Improved | None | >161K km | 1085 | 0.08715 | Pre-Euro |
| Truck/Bus | Diesel | Heavy | Direct Injection | Improved | None | >161K km | 1088 | 0.1899 | Pre-Euro |
| Truck/Bus | Diesel | Light | FI | Euro I | None | >161K km | 1118 | 0.04135 | Euro I |
| Truck/Bus | Diesel | Medium | FI | Euro I | None | >161K km | 1121 | 0.03505 | Euro I |
| Truck/Bus | Diesel | Heavy | FI | Euro I | None | >161K km | 1124 | 0.0764 | Euro I |
| Truck/Bus | Diesel | Light | FI | Euro II | None | 80-161K km | 1126 | 0.0304 | Euro II |
| Truck/Bus | Diesel | Medium | FI | Euro II | None | 80-161K km | 1129 | 0.0258 | Euro II |
| Truck/Bus | Diesel | Heavy | FI | Euro II | None | 80-161K km | 1132 | 0.0562 | Euro II |

**Table S-3.** Summary of EC/PM and OC/PM mass ratios sampled from literature

| No | EC/PM | OC/PM | Sampling method | Vehicle | Location | Component | Reference |
|---|---|---|---|---|---|---|---|
| 1 | 0.31±0.03 | 0.2±0.02 | Chassie dynamometer | Medium duty diesel truck | California, US | $PM_{2.5}$ | Schauer et al., 1999 |
| 2 | 0.34±0.04 | 0.25±0.12 | Chassie dynamometer | Heavy duty diesel vehicle | UK | Total PM | Shi et al., 2000 |
| 3 | 0.61±0.07 | 0.33±0.15 | On-road | Heavy duty diesel truck | California, US | $PM_{2.5}$ | Shah et al., 2004 |
| 4 | 0.44±0.09 | 0.33±0.15 | Tunnel | Mixed light & heavy duty vehicles | California, US | $PM_{2.5}$ | Gillies et al., 2001 |
| 5 | 0.43 | 0.33±0.15 | Tunnel | Light duty composite | Marseille, France | $PM_{2.5}$ | Haddad et al., 2009 |
| 6 | 0.45±0.48 | 0.22±0.23 | Tunnel | Mixed light & heavy duty vehicles | Guangzhou, China | $PM_{2.5}$ | He et al., 2008 |
| 7 | 0.46±0.23 | 0.2±0.11 | Chassie dynamometer | Light duty composite | Bangkok, Thailand | $PM_{2.5}$ | Oanh et al., 2010 |
| 8 | 0.48±0.18 | 0.13±0.14 | Chassie dynamometer | Heavy duty composite | Bangkok, Thailand | $PM_{2.5}$ | Oanh et al., 2010 |
| 9 | 0.5±0.57 | 0.18±0.14 | On-road | Light duty diesel cars | Delhi, India | $PM_{2.5}$ | Jaiprakash et al., 2015 |
| 10 | 0.48 | 0.25 | Chassie dynamometer | Light duty diesel cars | Taiwan | $PM_{2.5}$ | Chiang et al., 2012 |
| 11 | 0.5±0.44 | 0.26±0.28 | Tunnel | Mixed light & heavy duty vehicles | Hong Kong | $PM_{2.5}$ | Cheng et al., 2010 |

**Table S-2.** Parameters used for estimating fuel consumption for each type of brick kiln in the Kathmandu Valley

| Type of Kilns | Number of Kilns | Annual average production[a] (bricks/plant/year) | Monthly average production[b] Pj, (bricks/plant/month) | Average weight of a brick[c] $W_{brick}$ (kg per brick) | Specific energy consumption[d] $E_{brick}$ (MJ/kg-brick) | Specific energy density of coal $U_{coal}$, (MJ/kg-coal) | Coal consumed $BK_j$, kg-coal/plant/month |
|---|---|---|---|---|---|---|---|
| FCBTK | 46 | 5719626 | 953271 | 2.03 | 1.30 | 27 | 93173 |
| Hoffman | 2 | 20000000 | 3333333 | 2.03 | 1.36 | 27 | 340840 |
| VSBK | 1 | 8000000 | 1333333 | 2.03 | 0.80 | 27 | 80198 |
| Zigzag | 63 | 5719626 | 953271 | 2.03 | 1.03 | 27 | 73822 |

a: The annual average production of each type of kiln is obtained from http://doenv.gov.np/files/download/Report%20Brick%20Kiln%20%20Emission.pdf;

b: Nepal brick kilns usually operate 6 months per year, running from December to May;

c: Brick weight in Kathmandu Valley is 2.03 kg on average (CEN, 2009);

d. The value of specific energy consumption is obtained from http://www.ccacoalition.org/en/resources/factsheets-about-brick-kilns-south-and-south-east-asia

**Table S-1a.** Emission factors (g/kg fuel) for a zigzag kiln

| Compound (Formula) | Emission factors | Reference |
|:---:|:---:|:---:|
| EC | 0.1118 | |
| OC | 1.0577 | Jayarathne et al. (2018) |
| $SO_4$ | 4.8201 | |
| $PM_{2.5}$ | 15.11 | |
| Sulfur Dioxide ($SO_2$) | 12.7 | |
| Nitric Oxide (NO) | 1.28 | |
| Nitrogen Dioxide ($NO_2$) | $8.21 \times 10^{-2}$ | |
| Acetylene ($C_2H_2$) | $1.65 \times 10^{-2}$ | |
| Ethylene ($C_2H_4$) | $4.32 \times 10^{-2}$ | |
| Propylene ($C_3H_6$) | $6.58 \times 10^{-2}$ | |
| Methanol ($CH_3OH$) | 0.112 | |
| Formic Acid (HCOOH) | $5.84 \times 10^{-2}$ | |
| Acetic Acid ($CH_3COOH$) | 0.471 | |
| Phenol ($C_6H_5OH$) | $1.54 \times 10^{-2}$ | |
| 1,3-Butadiene ($C_4H_6$) | $1.51 \times 10^{-2}$ | |
| Isoprene ($C_5H_8$) | $2.46 \times 10^{-2}$ | |
| Nitrous Acid (HONO) | $4.45 \times 10^{-2}$ | |
| Methyl iodide ($CH_3I$) | $2.01 \times 10^{-3}$ | |
| 1,2-Dichloroethene ($C_2H_2Cl_2$) | $4.45 \times 10^{-5}$ | |
| Methyl nitrate ($CH_3NO_3$) | $2.92 \times 10^{-3}$ | |
| Ethane ($C_2H_6$) | $2.06 \times 10^{-3}$ | |
| Propane ($C_3H_8$) | $1.97 \times 10^{-3}$ | Stockwell et al. (2016) |
| i-Butane ($C_4H_{10}$) | $1.60 \times 10^{-3}$ | |
| n-Butane ($C_4H_{10}$) | $1.92 \times 10^{-3}$ | |
| 1-Butene ($C_4H_8$) | $1.68 \times 10^{-3}$ | |
| i-Butene ($C_4H_8$) | $1.47 \times 10^{-3}$ | |
| trans-2-Butene ($C_4H_8$) | $1.44 \times 10^{-3}$ | |
| cis-2-Butene ($C_4H_8$) | $9.65 \times 10^{-4}$ | |
| i-Pentane ($C_5H_{12}$) | $3.70 \times 10^{-2}$ | |
| n-Pentane ($C_5H_{12}$) | $3.26 \times 10^{-2}$ | |
| 1-Pentene ($C_5H_{10}$) | $1.60 \times 10^{-3}$ | |
| trans-2-Pentene ($C_5H_{10}$) | $2.64 \times 10^{-2}$ | |
| cis-2-Pentene ($C_5H_{10}$) | $9.01 \times 10^{-4}$ | |
| 3-Methyl-1-butene ($C_5H_{10}$) | $3.32 \times 10^{-4}$ | |
| 1,2-Propadiene ($C_3H_4$) | $2.15 \times 10^{-5}$ | |
| n-Hexane ($C_6H_{14}$) | $2.16 \times 10^{-2}$ | |
| n-Heptane ($C_7H_{16}$) | $3.04 \times 10^{-3}$ | |

**Table S-1b.** Emission factors (g/kg fuel) for a zigzag kiln

| Compound (Formula) | Emission factors | Reference |
|---|---|---|
| n-Octane ($C_8H_{18}$) | $1.58 \times 10^{-3}$ | |
| n-Nonane ($C_9H_{20}$) | $2.42 \times 10^{-3}$ | |
| n-Decane ($C_10H_{22}$) | $2.02 \times 10^{-3}$ | |
| 2,3-Dimethylbutane ($C_6H_{14}$) | $3.59 \times 10^{-3}$ | |
| 2-Methylpentane ($C_6H_{14}$) | $4.84 \times 10^{-3}$ | |
| 3-Methylpentane ($C_6H_{14}$) | $1.17 \times 10^{-2}$ | |
| 2,2,4-Trimethylpentane ($C_8H_{18}$) | $8.53 \times 10^{-4}$ | |
| Cyclopentane ($C_5H_{10}$) | $8.53 \times 10^{-4}$ | |
| Cyclohexane ($C_6H_{12}$) | $2.98 \times 10^{-3}$ | |
| Benzene ($C_6H_6$) | $8.25 \times 10^{-3}$ | |
| Toluene ($C_7H_8$) | $2.80 \times 10^{-2}$ | |
| Ethylbenzene ($C_8H_{10}$) | $1.35 \times 10^{-2}$ | |
| m/p-Xylene ($C_8H_{10}$) | $5.74 \times 10^{-2}$ | |
| o-Xylene ($C_8H_{10}$) | $2.18 \times 10^{-2}$ | |
| Styrene ($C_8H_8$) | $4.56 \times 10^{-3}$ | |
| i-Propylbenzene ($C_9H_{12}$) | $4.07 \times 10^{-4}$ | Stockwell et al. (2016) |
| n-Propylbenzene ($C_9H_{12}$) | $1.82 \times 10^{-3}$ | |
| 3-Ethyltoluene ($C_9H_{12}$) | $6.93 \times 10^{-3}$ | |
| 4-Ethyltoluene ($C_9H_{12}$) | $3.69 \times 10^{-3}$ | |
| 2-Ethyltoluene ($C_9H_{12}$) | $2.30 \times 10^{-3}$ | |
| 1,3,5-Trimethylbenzene ($C_9H_{12}$) | $4.30 \times 10^{-3}$ | |
| 1,2,4-Trimethylbenzene ($C_9H_{12}$) | $5.59 \times 10^{-3}$ | |
| 1,2,3-Trimethylbenzene ($C_9H_{12}$) | $2.03 \times 10^{-3}$ | |
| alpha-Pinene ($C_10H_{16}$) | $1.49 \times 10^{-3}$ | |
| beta-Pinene ($C_10H_{16}$) | $1.31 \times 10^{-3}$ | |
| Ethanol ($C_2H_{6O}$) | $4.84 \times 10^{-3}$ | |
| Acetaldehyde ($C_2H_{4O}$) | $6.94 \times 10^{-2}$ | |
| Acetone ($C_3H_{6O}$) | $1.46 \times 10^{-1}$ | |
| Butanal ($C_4H_{8O}$) | $2.19 \times 10^{-3}$ | |
| Butanone ($C_4H_{8O}$) | $2.29 \times 10^{-3}$ | |

**Table S0.** Composite emission factors of different vehicle types during running in the Kathmandu Valley, April 2015 (g/km) from the IVE model

| Vehicle types | CO | $SO_2$ | $NO_x$ | NMVOC | PM |
|---|---|---|---|---|---|
| Motorcycle | 7.78 | 0.01 | 0.20 | 2.21 | 0.11 |
| Bus/Minibus | 17.46 | 0.22 | 33.57 | 5.16 | 9.38 |
| Taxi | 37.72 | 0.04 | 1.61 | 4.69 | 0.01 |
| Car/Pickup | 3.72 | 0.06 | 2.47 | 0.77 | 0.28 |
| Van/Jeep | 3.97 | 0.14 | 4.51 | 0.62 | 1.03 |
| 3-wheeler | 6.86 | 3.14E-04 | 0.26 | 0.21 | 0.01 |
| Truck/Mini truck | 99.79 | 0.80 | 151.85 | 20.44 | 46.49 |

**Table S1.** Composite emission factors of different vehicle types during start-up in Kathmandu Valley, April 2015 (g/start) from the IVE model

| Vehicle types | CO | $SO_2$ | $NO_x$ | NMVOC | PM |
|---|---|---|---|---|---|
| Motorcycle | 9.11 | 0.00 | 1.91 | 1.91 | 0.12 |
| Bus/Minibus | 0.77 | 0.00 | 0.07 | 0.07 | 2.91 |
| Taxi | 27.60 | 0.00 | 3.07 | 3.07 | 0.01 |
| Car/Pickup | 6.20 | 0.00 | 0.51 | 0.51 | 0.11 |
| Van/Jeep | 1.93 | 0.00 | 0.15 | 0.15 | 0.27 |
| 3-wheeler | 4.00 | 3.92E-06 | 0.10 | 0.10 | 0.00 |
| Truck/Mini truck | 3.90 | 0.01 | 0.29 | 0.29 | 15.51 |

**Table S2.** Total daytime rainfall (mm) and average wind speed from 9:00 am - 18:00 pm during two episode periods.

| Date | 15-Apr | 16-Apr | 18-Apr | 19-Apr | 20-Apr | 21-Apr |
|---|---|---|---|---|---|---|
| Rainfall (mm) | 6.4 | 0 | 19.0 | 0 | 0 | 6.4 |
| Wind speed (m/s) | 1.5 | 2.4 | 2.3 | 2.7 | 3.9 | 4.3 |