# Peer review of "Nepal Ambient Monitoring and Source Testing Experiment (NAMaSTE): Emissions of particulate matter and sulfur dioxide from vehicles and brick kilns and their impacts on air quality in the Kathmandu Valley, Nepal"

_Atmospheric Chemistry and Physics, 2018_

## Referee Comment (RC1) · Anonymous Referee #1 · 8 Oct 2018

The air quality in Kathmandu Valley is evaluated in "Nepal Ambient Monitoring and Source Testing Experiment (NAMaSTE): Emissions of particulate matter and sulfur dioxide from vehicles and brick kilns and their impacts on air quality in the Kathmandu Valley, Nepal" paper by using an improved emissions inventory for road transport and brick kilns as input to a regional chemical transport model (WRF-Chem). Emissions estimation from road transport is based on the latest available data for vehicle registration and local emissions factors while for brick kilns the emissions were estimated

using measured emissions factors. This research provides to a better estimation of the impacts of emissions on air quality in Kathmandu, which is one of the most polluted city in Asia. The manuscript is well written and organised; however, to be published in ACP some additional explanations and corrections are needed.

General:

Scale concept should be introduced from the beginning since both emissions inventories and chemical transport models are built for either global, regional or local scale. For example, the relevance of nested model simulation for the Kathmandu Valley and its limitation for the specific conditions (e.g. orography) in this area should be discussed in section 2.1.

The study, among others, focuses on SO2. I would suggest a comparison of SO2 concentrations measured in Kathmandu Valley with different limit values (e.g. the limit value in European Union); this could be added on Figure 12.

Additional explanations should be provided for a better understanding of the validity of the comparison between pollutant concentrations from model simulations in 2015 and observation from 2013. Moreover, as input for the chemical transport model different emissions scenarios are used, i.e., HTAP for 2010 and emissions estimated in this study for the year 2015.

A section (e.g. 2.2.4 Emissions scenarios) about how the emissions scenarios were built is needed, including details about how HTAP emissions for Kathmandu Valley were derived; add a Table with emissions for each scenario. Please consult/add the following reference Li, M., Zhang, Q., Kurokawa, J., Woo, J.-H., He, K. B., Lu, Z., Ohara, T., Song, Y., Streets, D. G., Carmichael, G. R., Cheng, Y. F., Huo, H., Liu, F. Su, H., and Zheng, B.: MIX: a mosaic Asian anthropogenic emission inventory under the international collaboration framework of the MICS-Asia and HTAP, Atmos. Phys. Chem, 2017.

[Figure]

Since the brick kilns is missing in HTAP inventory, please delete from the Abstract and manuscript the statement "brick kilns account for nearly 70% of total sulfur dioxide (SO2) emissions from all sectors considered in HTAP_v2.2".

Clean up repetitive information throughout the text.

In section 5, the importance of this study for a future policy on emissions mitigation in this region should be highlighted.

Specific/Main text:

45 – please check/correct the values

120 – for clarity, please specify what was measured during the field campaign "NA-MaSTE" and what you compared

215 – please provide details about the observed surface SO2 concentrations at the monitoring stations in Kathmandu valley e.g. period/year

240 – replace "missions" with "emissions"

330 - please provide details about the observed EC concentrations at the monitoring stations in Kathmandu valley e.g. period/year

450 – please provided internet link/reference for "PANGAEA" and replace "will be available" with "are available"

Figures 5, 6 and 7 - please add legends

Figure 11 – are the Observations from April 2015?

Figure 12 - please provide explanation of the differences at "Bode" monitoring station

Supplementary Materials

Figure S2, caption – please delete "(HTAP_vehicle_brick)"

Table S1a, column "Age" – spell "K" out

Table S1a, column "Index" – provide the definition of the index

Table S1a, for line Truck/Bus Diesel Heavy FI Particulate/NOx None >161K km – please correct the information in the last four columns

Table S2 – please check the link http://www.ccacoalition.org/en/resources/factsheets-about-brick-kilns-south-and-south-east-a

Table S4, S5 – please provide references for the values in the tables.

---

## Referee Comment (RC2) · Anonymous Referee #2 · 27 Feb 2019

The concentration of different air-pollutants was simultaneously measured and evaluated using WRF-Chem in "Nepal Ambient Monitoring and Source Testing Experiment (NAMaSTE): Emissions of particulate matter and sulfur dioxide from vehicles and brick kilns and their impacts on air quality in the Kathmandu Valley, Nepal". Authors have done a non-trivial work by updating an existing emission inventory for Kathmandu Valley. However, there are some issues need to be resolved. 1. The authors have mentioned that "Since we lack survey data for trucks and cars in Kathmandu, we used the

data from Pune, India for these two types of vehicles" (Section 2.2.1). Using survey data of a western Indian city could increase the uncertainty related to the emission calculation. Authors, therefore, must include some logical arguments to establish the reasons behind using survey data of Pune in Kathmandu. 2. The authors have assumed the emitted PM as PM2.5 (section 2.2.1). Gillies et al. (2001) have estimated the emission factor of PM2.5-10 from a tunnel experiment in Los Angeles as 26% of total PM. Handler et al. (2008) reported the mass emission of PM2.5-10 almost equals to PM2.5 during an on-road motor-vehicular study in Vienna. Therefore, the authors need to explain the reason behind their assumptions logically. 3. The IVE model gives an output of PM10 (IVE model user manual, V2.0). As this model does not provide direct OC and EC output, therefore, the authors have used factors derived from Kim Onah et al. 2010. I would like to request the authors not to use the reference of Shresta et al. (2013) in this line. They should mention the reference of Kim Onah et al. 2010. The uncertainty related to this conversion factor for PM-to-EC and PM-to-OC is very high as shown by Kim Onah et al. (2010), and also the study has been carried out in a different country with different fuel quality and different meteorology compared to the present study. Therefore, I would like to suggest the authors use some probabilistic methods where the uncertainty related to these conversion factors could be taken care of. Else, the authors could include a separate section describing the uncertainty and if possible quantify it. 4. The authors have nicely explained the reasons behind the underestimation of EC. As per as the underestimation of SO2 is concerned, the authors have repeatedly discussed the Bode site. There is a distinct discrepancy between measured and observed values of SO2 in all the sites which indicates the presence of another source of SO2 that is not being considered. The authors need to rewrite the section (4.4) and try to explain the reasons behind the underestimation.

References

Gillies, J. A., Gertler, A. W., Sagebiel, J. C., & Dippel, N. W. (2001). On-road particulate matter (PM2. 5 and PM10) emissions in the Sepulveda Tunnel, Los Angeles,

California. Environmental science & technology, 35(6), 1054-1063. Handler, M., Puls, C., Zbiral, J., Marr, I., Puxbaum, H., & Limbeck, A. (2008). Size and composition of particulate emissions from motor vehicles in the Kaisermühlen-Tunnel, Vienna. Atmospheric Environment, 42(9), 2173-2186. Oanh, N. T. K., Thiansathit, W., Bond, T. C., Subramanian, R., Winijkul, E., & Paw-armart, I. (2010). Compositional characterization of PM2. 5 emitted from in-use diesel vehicles. Atmospheric Environment, 44(1), 15-22.

————————————————————

---

## Author Comment (AC1) · 10 Apr 2019

**Response to reviewers' comments**

We thank the Editor and the reviewer for the constructive comments. We have addressed all the comments, and listed our point-by-point reply below. We list the reviewers' comments in black and our replies in blue.

The air quality in Kathmandu Valley is evaluated in "Nepal Ambient Monitoring and Source Testing Experiment (NAMaSTE): Emissions of particulate matter and sulfur dioxide from vehicles and brick kilns and their impacts on air quality in the Kathmandu Valley, Nepal" paper by using an improved emissions inventory for road transport and brick kilns as input to a regional chemical transport model (WRF-Chem). Emissions estimation from road transport is based on the latest available data for vehicle registration and local emissions factors while for brick kilns the emissions were estimated using measured emissions factors. This research provides to a better estimation of the impacts of emissions on air quality in Kathmandu, which is one of the most polluted city in Asia. The manuscript is well written and organized; however, to be published in ACP some additional explanations and corrections are needed.

General: Scale concept should be introduced from the beginning since both emissions inventories and chemical transport models are built for either global, regional or local scale. For example, the relevance of nested model simulation for the Kathmandu Valley and its limitation for the specific conditions (e.g. orography) in this area should be discussed in section 2.1.

We added the following sentences in section 2.1, line 108:
*The topography of the innermost model domain is complicated, with the Himalayas range sitting across west to east and separating the Indian subcontinent from the Tibetan Plateau. Even when we use 3 km spacing for the nested domain, the model is unable to resolve the very steep topographic features but this was the best we could do with this project, given the resolution of emissions available.*

The study, among others, focuses on $SO_2$. I would suggest a comparison of $SO_2$ concentrations measured in Kathmandu Valley with different limit values (e.g. the limit value in European Union); this could be added on Figure 12.

We added the WHO guideline for 24-h mean $SO_2$ concentrations (20 µg m$^{-3}$) on Figure 12 and we added the following to the manuscript in section 4.4, line 432:
*None of these sites exceeded the Nepal national air quality standard of 70 µg m$^{-3}$ for 24 h mean, but $SO_2$ concentrations at Bode site were almost twice as high as the WHO standard of 20 µg m$^{-3}$.*

[Figure]

Additional explanations should be provided for a better understanding of the validity of the comparison between pollutant concentrations from model simulations in 2015 and observation from 2013.

The $SO_2$ measurements in the Kathmandu Valley are not routinely performed, as done in many other urban cities. We found limited amount of observational data for ambient $SO_2$. Our own NAMaSTE campaign only collected daily $SO_2$ at the Bode site but we found that Kiros et al. (2016) reported their measurements between March 23 and May 18, 2013. The two-week mean $SO_2$ concentration from the NAMaSTE in 2015 was 39.7 µg m$^{-3}$ at Bode, while the 8-week mean in 2013 by Kiros et al. (2016) was 39.2 µg m$^{-3}$. Because we wanted to highlight the difference in $SO_2$ concentrations at the Bode site compared to others, we found that this could be potentially helpful in illustrating the magnitude difference in observational data within the Kathmandu Valley. We have included the following sentences in Section 4.4, Line 434:

*Since our own NAMaSTE campaign only collected $SO_2$ at the Bode site, we also included the study of Kiros et al. (2016) to illustrate the magnitude difference in observational data at different locations within the Kathmandu Valley. The two-week mean $SO_2$ concentration from the NAMaSTE in 2015 was 39.7 µg m$^{-3}$ at Bode, while the 8-week mean in 2013 by Kiros et al. (2016) was 39.2 µg m$^{-3}$, showing similarities, giving us confidence that comparing the magnitude difference was possible, despite the difference in observed years.*

Moreover, as input for the chemical transport model different emissions scenarios are used, i.e., HTAP for 2010 and emissions estimated in this study for the year 2015.

We added the following sentence in the new Section 2.2.1 Emission Scenarios in 2015, line 142:
*We used the latest available HTAP_v2.2 for 2010 as the baseline inventory, as this was the closest year to 2015 that we had the data for at that time of model simulation. The vehicle and brick kiln emissions were developed for year 2015.*

A section (e.g. 2.2.4 Emissions scenarios) about how the emissions scenarios were built is needed, including details about how HTAP emissions for Kathmandu Valley were derived; add a Table with emissions for each scenario. Please consult/add the following reference Li, M., Zhang, Q., Kurokawa, J., Woo, J.-H., He, K. B., Lu, Z., Ohara, T., Song, Y., Streets, D. G., Carmichael, G. R., Cheng, Y. F., Huo, H., Liu, F. Su, H., and Zheng, B.: MIX: a mosaic Asian anthropogenic emission inventory under the international collaboration framework of the MICS-Asia and HTAP, Atmos. Phys. Chem, 2017.

We have added the new section 2.2.1. named *Emissions Scenarios in 2015* to describe the details:
*We used three emissions scenarios (Table 1) to investigate the impact of emissions on local air quality in the Kathmandu Valley. The first emissions scenario is the same as the original HTAP_v2.2 (Janssens-Maenhout et al. 2015). HTAP is a gridded global emission inventory combined with the regional inventories and gap-filled with the Emissions Database for Global Atmospheric Research (EDGAR v4.3) (Janssens-Maenhout et al. 2013). In Asia, HTAP_v2.2 uses MIX inventory, a regional emission inventory in Asia, which is also developed based on the 'mosaic' approach including multiple existing national inventories (Li et al. 2017). The second emissions scenario utilizes the original HTAP_v2.2 with updated vehicle emissions (Section 2.2.1). The third scenario is built on the second scenario and adding emissions from brick kilns (Section 2.2.2). We used the latest available HTAP_v2.2 for 2010 as the baseline inventory, as this is the closest year to 2015 that we have the data for. The vehicle and brick kiln emissions were developed for year 2015.*

*Table 1 Description for each emissions scenario in 2015*

| Emissions Scenarios | Description |
| --- | --- |
| HTAP | Original HTAP_v2.2 emission inventory |
| HTAP_vehicle | Original HTAP_v2.2 + updated vehicle emissions |
| HTAP_vehicle_brick | Original HTAP_v2.2 + updated vehicle emissions + brick kiln emissions |

Since the brick kilns is missing in HTAP inventory, please delete from the Abstract and manuscript the statement "brick kilns account for nearly 70% of total sulfur dioxide (SO2) emissions from all sectors considered in HTAP_v2.2". Clean up repetitive information throughout the text.

We have rewritten the sentence as follows:
*HTAP_v2.2 does not include brick sector and we find that our sulfur dioxide (SO$_2$) emissions estimates from brick kilns to be comparable to 70% of the total SO$_2$ emissions considered in HTAP_v2.2.*

In section 5, the importance of this study for a future policy on emissions mitigation in this region should be highlighted.

We have added the following sentence in Section 5, line 495, to highlight the importance of our study based on the reviewer's constructive suggestion:
*A more comprehensive and accurate emission inventory allows the local government to identify and define key emission sources in the Kathmandu Valley. The improved emission inventory is urgently needed to robustly evaluate the effectiveness of various future policies on emission mitigation in this region.*

Specific/Main text:
45 – please check/correct the values

It has been corrected. The sentence reads as:
*In 2010, the annual emissions of EC and OC from diesel-powered vehicles in the Kathmandu Valley were estimated at 2,117 and 570 ton/year, respectively.*

120 – for clarity, please specify what was measured during the field campaign "NAMaSTE" and what you compared

The field campaign provided emission factors of various air pollutants from brick kilns. We also obtained meteorological data, EC concentrations, and SO$_2$ concentrations at Bode site from this campaign. We compared the above observational data to the results from the three simulations with different emissions inputs: HTAP, HTAP with updated vehicle emissions, and HTAP with updated vehicle emissions plus brick kiln emissions. We have a section 2.3 devoted to explaining this and we clarified by referring to this specific section, as follows in section in 2.1, line 124:
*We conducted each simulation for the two week period of April 12-24, 2015 during which observational data from the NAMaSTE field campaign were available for comparison (Section 2.3).*

215 – please provide details about the observed surface SO2 concentrations at the monitoring stations in Kathmandu valley e.g. period/year

The following descriptions have been added in section 2.3, line 242:
*The observed surface SO$_2$ is 8-week mean concentrations between March 23 and May 18, 2013 from Kiros et al. (2016). They were measured at 15 sites in the valley, including five urban sites (Bode, Indrachowk, Maharajgunj, Mangal Bazaar, Suryabinayak), four suburban sites (Bhaisepati, Budhanilkantha, Kirtipur,*

*Lubhu), and six rural sites (Bhimdhunga, Nagarkot, Naikhandi, Nala Pass, Sankhu, Tinpiple) (Kiros et al., 2016).*

240 – replace "missions" with "emissions"

Corrected.

330 - please provide details about the observed EC concentrations at the monitoring stations in Kathmandu valley e.g. period/year

We have added the sampling dates in Line 241 in the manuscript:
*Concentrations of daily EC and $SO_2$ at Bode, Kathmandu were sampled at a height of 20 m during the NAMaSTE field campaign in April 12-24, 2015.*

450 – please provided internet link/reference for "PANGAEA" and replace "will be available" with "are available"

The data will be available upon acceptance to ACP in the PANGAEA database.

Figures 5, 6 and 7 - please add legends

Added.

Figure 11 – are the Observations from April 2015?

Yes, the observed EC concentrations were obtained in April 2015 from the NAMaSTE campaign. We have added a sentence to make that clear in the Figure caption:
*Figure 11. Comparisons of observed (blue dots) and modeled EC concentrations in daytime, nighttime, and daily mean at Bode. Observed values are taken during the NAMaSTE campaign (Jayarathne et al., 2018)*

Figure 12 - please provide explanation of the differences at "Bode" monitoring station Supplementary Materials

We have added the following to clarify the differences at Bode station in section 4.4, line 443:
*This underestimation is probably due to brick kiln $SO_2$ emissions. We applied an emission factor of 12.7 g/kg of fuel measured from zigzag kilns (Stockwell et al, 2016) to all types of brick kilns. This was the only available observational data in Nepal at the time of this study. A more recent study by Nepal et al. (2019) reported that the mean value of $SO_2$ emission factor from zigzag kilns is $24\pm22$ g/kg fuel, which is almost twice as high as that used in our study. If we doubled our $SO_2$ emissions for brick kilns, the modeled $SO_2$ concentrations would be much closer to the observations. Assuming the linear relationship in $SO_2$, the average difference between the observed and modeled $SO_2$ concentrations would drop from 4.4 µg $m^{-3}$ to 2.8 µg $m^{-3}$. We plan to revisit our brick kiln emissions inventory, as more emission factors become available. Our study highlights the importance of improving emission factor of $SO_2$ for brick kilns in Nepal.*

Figure S2, caption – please delete "(HTAP_vehicle_brick)"

It has been deleted.

Table S1a, column "Age" – spell "K" out

We have added the unit of Age as "km travelled" and deleted 'K'.

Table S1a, column "Index" – provide the definition of the index Table S1a, for line Truck/Bus Diesel Heavy FI Particulate/NOx None >161K km – please correct the information in the last four columns

We added the following note in Table S1a. The line *Truck/Bus Diesel Heavy FI Particulate/NOx None* has been corrected.
*Index is a serial of numbers from 0 to 1371 used to label the 1371 types of vehicle technologies in the IVE model.*

Table S2 – please check the link http://www.ccacoalition.org/en/resources/factsheetsabout-brick-kilns-south-and-south-east-a

The link has been corrected. http://www.ccacoalition.org/en/resources/factsheets-about-brick-kilns-south-and-south-east-asia

Table S4, S5 – please provide references for the values in the tables.

Tables S4 & S5 (Tables S5 & S6 in the revised manuscript) were created using the IVE model and we added that these were the results of the IVE model in the captions.

References
Janssens-Maenhout G, Pagliari V, Guizzardi D, Muntean, M. 2013. Global emission inventories in the emission database for global atmospheric research (EDGAR) – manual (i): Gridding: EDGAR emissions distribution on global gridmaps. JRC report. JRC Report EUR 25785 EN, ISBN 978-92-79-28283-6, doi:10.2788/81454.
Janssens-Maenhout G, Crippa M, Guizzardi D, Dentener F, Muntean M, Pouliot G, et al. 2015. Htap_v2.2: A mosaic of regional and global emission grid maps for 2008 and 2010 to study hemispheric transport of air pollution. Atmos Chem Phys 15:11411-11432.
Li M, Zhang Q, Kurokawa JI, Woo JH, He K, Lu Z, et al. 2017. Mix: A mosaic asian anthropogenic emission inventory under the international collaboration framework of the MICS-ASIA and HTAP, *Atmospheric Chemistry and Physics*, *17*, 935-963.
Nepal S, Mahapatra PS, Adhikari S, Shrestha S, Sharma P, Shrestha K, Banmali Pradhan, Bidya Praveen, P. S. (2019). A comparative study of stack emissions from straight-line and zigzag brick kilns in Nepal, *Atmosphere*, **10** (3), 107.

---

## Author Comment (AC2) · 10 Apr 2019

**Response to reviewers' comments**

We thank the Editor and the reviewer for the constructive comments. We have addressed all the comments, and listed our point-by-point reply below. We list the reviewers' comments in black and our replies in blue.

1. The concentration of different air-pollutants was simultaneously measured and evaluated using WRF-Chem in "Nepal Ambient Monitoring and Source Testing Experiment (NAMaSTE): Emissions of particulate matter and sulfur dioxide from vehicles and brick kilns and their impacts on air quality in the Kathmandu Valley, Nepal". Authors have done a non-trivial work by updating an existing emission inventory for Kathmandu Valley. However, there are some issues need to be resolved. 1. The authors have mentioned that "Since we lack survey data for trucks and cars in Kathmandu, we used the data from Pune, India for these two types of vehicles" (Section 2.2.1). Using survey data of a western Indian city could increase the uncertainty related to the emission calculation. Authors, therefore, must include some logical arguments to establish the reasons behind using survey data of Pune in Kathmandu.

We agree with the reviewer that the survey data of Pune from India could introduce additional uncertainty compared to using the local data of Kathmandu. We only did this because the data for Kathmandu was unfortunately not available. Pune was the only representative city within South Asia, where the International Sustainable System Research Center (ISSRC) conducted a detailed study of vehicle activity. We added the following in the manuscript in section 2.2.2 in line 164:
*Pune was the only representative city within South Asia, where the International Sustainable System Research Center (ISSRC) conducted a detailed study of vehicle activity.*

2. The authors have assumed the emitted PM as $PM_{2.5}$ (section 2.2.1). Gillies et al. (2001) have estimated the emission factor of $PM_{2.5-10}$ from a tunnel experiment in Los Angeles as 26% of total PM. Handler et al. (2008) reported the mass emission of $PM_{2.5-10}$ almost equals to $PM_{2.5}$ during an on-road motor-vehicular study in Vienna. Therefore, the authors need to explain the reason behind their assumptions logically. The IVE model gives an output of $PM_{10}$ (IVE model user manual, V2.0)

The reviewer correctly points out that the PM in the IVE model refers to $PM_{10}$. The ratio of $PM_{2.5}$ to $PM_{10}$ is 0.92 for diesel vehicles and 0.88 for gasoline vehicles according to EPA's 2014 MOVES model. Since $PM_{2.5}$ is the dominant particulate matter and diesel engines appear to be the biggest source of $PM_{2.5}$ on road, we assumed the emitted PM as $PM_{2.5}$ to simplify our estimation. Both Gillies et al. (2001) and Handler et al. (2008) indicated that emissions of coarse particles were dominated by resuspended dust as well as by brake wear, while fine particles were mainly derived from combustion processes. We added the following in the new section 2.2.2 in l. 174:
*All emitted PM was assumed to be $PM_{2.5}$ because the ratio of $PM_{2.5}$ to $PM_{10}$ is 0.92 for diesel vehicles and 0.88 for gasoline vehicles in EPA 2014 MOVES model. Studies such as Gillies et al. (2001) and Handler et a. (2008) have also found that 74% and 67% of $PM_{10}$ is $PM_{2.5}$ in on-road studies. Although we understand that assuming all emitted $PM_{10}$ to be $PM_{2.5}$ is potentially an overestimation, we believe that this is acceptable, given the lack of observational data in Nepal or in South Asia.*

3. As this model does not provide direct OC and EC output, therefore, the authors have used factors derived from Kim Onah et al. 2010. I would like to request the authors not to use the reference of Shresta et al.(2013) in this line. They should mention the reference of Kim Onah et al. 2010. The uncertainty related to this conversion factor for PM-to-EC and PM-to-OC is very high as shown by Kim Onah et al. (2010), and also the study has been carried out in a different country with different fuel quality and different meteorology compared to the present study. Therefore, I would like to suggest the authors use some probabilistic methods where the uncertainty related to these conversion factors could be taken care of. Else, the authors could include a separate section describing the uncertainty and if possible quantify it.

We have changed the reference as suggested by the reviewer. In addition, we created a pool of PM-to-EC and PM-to-OC conversion factors from multiple references to describe the potential uncertainties of these conversion factors (new Table S1). We also corrected our description and briefly discussed the uncertainties. The revised sentences read as follows in section 2.2.2, l. 180:

*Because the IVE model does not directly estimate emissions of EC or OC, we used conversion factors derived from the study of Kim Oanh et al. (2010) to estimate these emissions. Kim Oanh et al. (2010) specifically focused on the emissions of diesel vehicles in developing countries and had tested a large number of vehicles. For vans, we used EC/PM mass ratio of 0.46 and OC/PM of 0.2, while for trucks and buses, we used EC/PM of 0.48 and OC/PM of 0.13. We collected a group of these conversion factors from different studies in Table S1. Our EC/PM mass ratio is close to median value of all the studies listed below. While we acknowledge that using conversion factors from one study ignores the potential uncertainty due to driving pattern, weather conditions, fuel quality, and vehicle characteristics, we also feel that our estimate provides a good middle ground, given the existing study results.*

Table S1: Summary of EC-to-PM and OC-to-PM mass ratios sampled from literature

| No. | EC/PM | OC/PM | Sampling method | Vehicle | Location | Component | Reference |
|---|---|---|---|---|---|---|---|
| 1 | $0.31 \pm 0.03$ | $0.2 \pm 0.02$ | Chassie dynamometer | Medium duty diesel truck | California, US | $PM_{2.5}$ | Schauer et al., 1999 |
| 2 | $0.34 \pm 0.04$ | $0.25 \pm 0.12$ | Chassie dynamometer | Heavy duty diesel vehicle | UK | Total PM | Shi et al., 2000 |
| 3 | $0.61 \pm 0.07$ | $0.33 \pm 0.15$ | On-road | Heavy duty diesel truck | California, US | Total PM | Shah et al., 2004 |
| 4 | $0.44 \pm 0.09$ | $0.33 \pm 0.15$ | Tunnel | Mixed light & heavy duty vehicles | California, US | $PM_{2.5}$ | Gillies et al., 2001 |
| 5 | 0.43 | 0.16 | Tunnel | Light duty composite | Marseille, France | $PM_{2.5}$ | Haddad et al., 2009 |
| 6 | $0.45 \pm 0.48$ | $0.22 \pm 0.23$ | Tunnel | Mixed light & heavy duty vehicles | Guangzhou, China | $PM_{2.5}$ | He et al., 2008 |
| 7 | $0.46 \pm 0.23$ | $0.2 \pm 0.11$ | Chassie dynamometer | Light duty composite | Bangkok, Thailand | $PM_{2.5}$ | Kim Oanh et al., 2010 |
| 8 | $0.48 \pm 0.18$ | $0.13 \pm 0.14$ | Chassie dynamometer | Heavy duty composite | Bangkok, Thailand | $PM_{2.5}$ | Kim Oanh et al., 2010 |
| 9 | $0.5 \pm 0.57$ | $0.18 \pm 0.14$ | On-road | Light duty diesel cars | Deli, India | $PM_{2.5}$ | Jaiprakash, 2015 |
| 10 | 0.48 | 0.25 | Chassie dynamometer | Light duty diesel cars | Taiwan | $PM_{2.5}$ | Chiang et al., 2012 |
| 11 | $0.5 \pm 0.44$ | $0.26 \pm 0.28$ | Tunnel | Mixed light & heavy duty vehicles | Hong Kong | $PM_{2.5}$ | Cheng et al., 2010 |

4. The authors have nicely explained the reasons behind the underestimation of EC. As per as the underestimation of $SO_2$ is concerned, the authors have repeatedly discussed the Bode site. There is a distinct discrepancy between measured and observed values of $SO_2$ in all the sites which indicates the presence of another source of $SO_2$ that is not being considered. The authors need to rewrite the section (4.4) and try to explain the reasons behind the underestimation.

Based on our analysis, brick kilns are one of the largest sources of $SO_2$ in Kathmandu Valley, which has not been considered in current global or regional emission inventories. Since we have limited emission factors for $SO_2$ from brick kilns, we hypothesized that the main reason was due to our underestimation of $SO_2$ emissions from this sector, although it was our best estimate. We added the following sentence to discuss this issue in section 4.4, l. 449:
*This underestimation is probably due to brick kiln $SO_2$ emissions. We applied an emission factor of 12.7 g/kg of fuel measured from Zigzag kilns (Stockwell et al, 2016) to all types of brick kilns. This was the only available observational data in Nepal at the time of this study. A more recent study by Nepal et al. (2019) reported that the mean value of $SO_2$ emission factor from Zigzag kilns is $24\pm22$ g/kg fuel, which is almost twice as high as that used in our study. If we doubled our $SO_2$ emissions for brick kilns, the modeled $SO_2$ concentrations would be much closer to the observations. Assuming the linear relationship in $SO_2$, the average difference between the observed and modeled $SO_2$ concentrations would drop from 4.4 µg m$^{-3}$ to*

*2.8 μg m$^{-3}$. We plan to revisit our brick kiln emissions inventory, as more emission factors become available. Our study highlights the importance of improving emission factor of SO$_2$ for brick kilns in Nepal."*

References

Gillies, J. A., Gertler, A. W., Sagebiel, J. C., & Dippel, N. W. (2001). On-road particulate matter (PM$_{2.5}$ and PM$_{10}$) emissions in the Sepulveda Tunnel, Los Angeles, California, *Environmental Science & Technology*, **35** (6), 1054-1063.

Handler, M., Puls, C., Zbiral, J., Marr, I., Puxbaum, H., & Limbeck, A. (2008). Size and composition of particulate emissions from motor vehicles in the Kaisermühlen-Tunnel, Vienna, *Atmospheric Environment*, **42** (9), 2173-2186.

Kim Oanh, N. T., Thiansathit, W., Bond, T. C., Subramanian, R., Winijkul, E., & Paw-armart, I. (2010). Compositional characterization of PM$_{2.5}$ emitted from in-use diesel vehicles. *Atmospheric Environment*, **44** (1), 15-22.

Nepal S, Mahapatra PS, Adhikari S, Shrestha S, Sharma P, Shrestha K, Banmali Pradhan, Bidya Praveen, P. S. (2019). A comparative study of stack emissions from straight-line and zigzag brick kilns in Nepal, *Atmosphere*, **10** (3), 107.